# A Deep Latent Space Model for Directed Graph Representation Learning

## Abstract

Graph representation learning is a fundamental problem for modeling relational data and benefits a number of downstream applications. Traditional Bayesian-based random graph models and recent deep learning based methods are complementary to each other in interpretability and scalability. To take the advantages of both models, some combined methods have been proposed. However, existing models are mainly designed for *undirected graphs*, while a large portion of real-world graphs are directed. The focus of this paper is on the more challenging *directed graphs* where both the existences and directions of edges need to be learned. We propose a Deep Latent Space Model (DLSM) for directed graphs to incorporate the traditional latent space random graph model into deep learning frameworks via a hierarchical variational auto-encoder architecture. To adapt to directed graphs, our model generates multiple highly interpretable latent variables as node representations, and the interpretability of representing node influences is theoretically proved. The experimental results on real-world graphs demonstrate that our proposed model achieves the state-of-the-art performances on link prediction and community detection tasks while generating interpretable node representations of directed graphs.

## 1 Introduction

Learning representations on graphs is a fundamental problem for graph analysis and benefits a number of downstream applications, typically including link prediction, community detection and influential node identification. Traditionally, a plethora of Bayesian-based random graph models have been proposed for learning graph representations (Holland et al., 1983; Hoff et al., 2002; Karrer & Newman, 2011; Sewell & Chen, 2015). Despite the ideal interpretability of these methods, they are unscalable to model large-scale graphs due to the expensive iterative inference procedures. Boosted by the powerful representation learning ability, graph neural networks (GNNs) have been proposed to learn the topology of graph-structured data (Kipf & Welling, 2017; Hamilton et al., 2017; Veličković et al., 2018; Zeng et al., 2020). However, these GNN-based methods usually bring about the interpretability issues, which restrict the practical applications of these methods.

As an effective deep learning method based on the Bayesian theory, the variational auto-encoders (VAEs) (Kingma & Welling, 2014b) have been proposed for representation learning on graph-structured data (Kipf & Welling, 2016; Grover et al., 2019; Salha et al., 2019). To capture the mutual facilitation of random graph models and deep learning based methods, some recent research attempts to combine them using VAEs, and build the deep generative random graph models (Mehta et al., 2019; Sarkar et al., 2020). However, most of these VAE-based methods are designed only for *undirected graphs*. In contrast, a large portion of real-world graphs, such as social and citation networks, are *directed graphs*, of which undirected graphs are special cases (i.e. an undirected edge can be regarded as a bidirectional edge). The representation learning on directed graphs is particularly challenging since not only the existences but also the directions of edges need to be learned. Although directed graphs have been extensively studied by traditional research on random graph models (Hoff et al., 2002; Krivitsky et al., 2009; Sewell & Chen, 2015), there are still few deep learning based methods and no combined models existed to address them.

In this paper, we present the first VAE-based generative model for directed graphs and propose the Deep Latent Space Model (DLSM), which combines graph convolutional networks (GCNs) with the

Bayesian random graph model using a hierarchical VAE architecture. Within our model, a deep GCN is leveraged to encode the asymmetric adjacency matrix and node features as hidden states, which are layer-wise transformed to variational parameters to generate latent random variables as interpretable node representations. Then, the hierarchical decoder network reconstructs the adjacency matrix using the learned representations.

To better capture the characteristics of graphs for deep random graph models, our method generates three types of interpretable representations, including the latent positions, node random factors and community memberships. The latent positions measure the distances between nodes and enable our method to model the *link reciprocity* of directed graphs, which means that both bidirectional and unidirectional edges exist. To characterize the *degree heterogeneity*, we introduce a pair of node random factors, namely the social activity and popularity, thus the asymmetric structure of directed edges can be modeled. Besides, we use a binary latent variable, named the community membership, to accommodate an overlapping *community structure*. We further prove that these node-specific variables, including the latent positions and random factors, are interpretable for representing the existence and strength of node influences.

For fast inference, we leveraged the stochastic gradient variational Bayes (SGVB) algorithm (Kingma & Welling, 2014a), which is far more efficient and scalable compared to the traditional iterative inference methods. We conduct experiments on link prediction task to verify the effectiveness of directed graph modeling with several real-world graphs, and further demonstrate the interpretability of latent variables on community detection task.

The main contributions of this work are summarized as follows:

1. We propose the DLSM model for directed graphs to incorporate the latent space model into deep learning frameworks using a hierarchical VAE architecture, which can generate highly interpretable representations of graph nodes for multiple downstream tasks.

2. To adapt our method to directed graphs, we introduce the social activity and popularity factors of each node, with theoretical proof of their capabilities in representing node influences.

3. We conduct experiments on five real-world graph datasets, and the results show that our proposed method can well support downstream tasks and achieve the state-of-the-art performances in link prediction and community detection.

## 2 RELATED WORK

In this section, we briefly review the traditional Bayesian-based random graph models and deep learning based generative graph representation methods.

### 2.1 BAYESIAN-BASED RANDOM GRAPH MODELS

Classic Bayesian-based random graph models have developed for decades and are still valued in modeling and generating relational data. This methods regard graph nodes as random variables following a prior distribution, and learns low-dimensional embeddings via the posterior distributions of nodes. One of the most well-known methods is the stochastic blockmodel (SBM) (Holland et al., 1983), which generates a latent variable indicating the community membership of each node. Following SBM, a large number of variants have been proposed, such as allowing an overlapping community structure (Miller et al., 2009) and involving degree heterogeneity (Karrer & Newman, 2011).

Another important random graph model is the latent space model (LSM) (Hoff et al., 2002), which assumes each node as a position in an unobservable latent space and use distances to measure the relationships between nodes. Such method has soon been extended to model directed graphs by involving the degree heterogeneity of nodes (Krivitsky et al., 2009; Sewell & Chen, 2015). Although these above methods provide good theoretical properties and learn interpretable node embeddings, they rely on either the MCMC posterior sampling or mean-field variational inference with dramatically high computational complexity, and thus are usually unfeasible when modeling large-scale graphs.

## 2.2 DEEP LEARNING BASED MODELS FOR UNDIRECTED GRAPHS

The intriguing achievements of deep learning models for learning representations of Euclidean data have encouraged efforts to employ them on graph-structured data. The earliest attempts include DeepWalk (Perozzi et al., 2014) and node2vec (Grover & Leskovec, 2016), both of which encode local relations as node representations by conducting random walks on a graph.

**GNN-based graph models** More recently, many GNN-based models have been proposed for graph representation learning. Kipf & Welling (2017) first leverages the spectral GCN to learn node embeddings using the global topology of a graph. GraphSAGE (Hamilton et al., 2017) samples a fixed-size neighborhood of each node for the inductive representation learning, which allows unseen nodes to be excluded during training. The graph attention networks (Veličković et al., 2018) introduce the attention mechanism to aggregate the information from neighbors. GraphSAINT (Zeng et al., 2020) constructs minibatches by sampling subgraphs across GCN layers. (Zhang et al., 2021a) proposes a multi-node representation learning method for link prediction using the node labeling trick. These methods usually learn representations for a single or a set of nodes and only focus on some specific task of graph modeling, such as node classification or link prediction.

**Deep generative graph models** Another stream of research is the deep generative models, which attempt to generate the full structures of graphs and are typically able to be applied for multiple downstream tasks. These methods usually consist of a inferential model to capture the inner probabilistic distribution of graphs and an unsupervised generative model to generate the similar data. The variational graph auto-encoders (VGAE) (Kipf & Welling, 2016), for example, sample node embeddings from variational Normal distributions to reconstruct the adjacency matrix. GraphGAN Wang et al. (2018) unifies a generator and a discriminator to learn node embeddings by playing a minimax game. GraphRNN You et al. (2018) generates small graphs through a sequence of node and edge formations. Recently, some work tries to incorporate traditional random graph models into deep learning frameworks. Mehta et al. (2019) first combines the classic SBM with GCN to learn sparse node embeddings. Furthermore, Sarkar et al. (2020) builds a Gamma ladder VAE (Sønderby et al., 2016) architecture to discover the communities at multiple levels of granularities.

## 2.3 DEEP LEARNING BASED MODELS FOR DIRECTED GRAPHS

Despite a plethora of literature for undirected graph modeling, the deep learning based methods for directed graphs are still relatively rare. Existing methods are mostly designed for some special cases and are not generalizable enough. For instance, Salha et al. (2019) and Zhang et al. (2021b) focus on the unidirectional graphs and neglect the link reciprocity of generalized directed graphs. Funke et al. (2020) learns the low-dimensional statistical manifold embedding for unsupervised directed graph modeling. Zhu et al. (2021) learns a pair of source and target embeddings for each node using the generative adversarial network. In this paper, we propose a novel deep generative model and learn interpretable node representations for generalized directed graphs where both uni- and bi-directional edges exist.

## 3 DEEP LATENT SPACE MODEL

Consider a directed graph containing $n$ nodes. The input data include an asymmetric adjacency matrix $\mathbf{A} = (\mathbf{A}_{ij}) \in \{0,1\}^{n \times n}$, where each binary element $\mathbf{A}_{ij}$ denotes the presence (1) or not (0) of the directed edge from node $i$ to $j$, and a node attribute matrix $\mathbf{X} \in \mathbb{R}^{n \times p}$. Throughout this paper we assume all edges to be conditionally independent and satisfy $\mathbf{A}_{ij}|\Theta \sim \text{Bernoulli}(p_{ij})$, where $\Theta$ is the collection of latent variables (representations) and $p_{ij} \in (0, 1)$ is the posterior probability to form an edge. The objective is to learn the representations which can best reconstruct the adjacency matrix.

We propose a deep VAE architecture composed of a GCN encoder and a hierarchical latent space model (HLSM) decoder, as shown in Fig. 1. The encoder block takes the adjacency $\mathbf{A}$ and attribute matrices $\mathbf{X}$, if available, as inputs and learns a hidden state for each node using a directed GCN. Then, the decoder block recursively generates interpretable latent variables from variational posteriors via Monte Carlo sampling and reconstructs the adjacency matrix as outputs. For inference, the

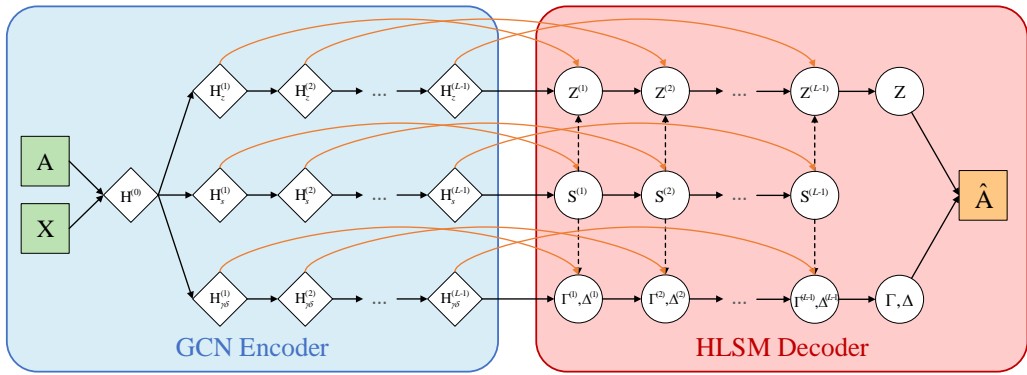

Figure 1: The architecture of hierarchical variational auto-encoder. The HLSM decoder generates a adjacency matrix with latent variables, which are sparsified by the community membership (black dashed arrows). The GCN encoder learns variables using not only the feedforward priors from previous layers (black solid arrows), but also the skipping likelihood of the inputs (orange arcs).

hierarchical architecture enables the model to learn variables using not only the priors passed from previous layers of the decoder, but also the approximate likelihood learned by the encoder.

### 3.1 HLSM DECODER

The HLSM decoder learns interpretable node representations based on the classic LSM approach and generates an adjacency matrix. Assume each node to correspond to a latent position in an unobservable $D$-dimensional space, denoted as $z_i \in \mathbb{R}^D$. The distances between latent positions indicate the relationships of nodes. The closer two latent positions are, the more possible an edge will exist. Formally, an directed edge is generated by following

$$\mathrm{P}\left(\mathbf{A}_{ij}=1|\mathbf{X},\Theta\right)=\sigma\left(\beta_0-\beta_{out}\|\boldsymbol{\gamma}_i\odot(\boldsymbol{z}_i-\boldsymbol{z}_j)\|-\beta_{in}\|\boldsymbol{\delta}_j\odot(\boldsymbol{z}_i-\boldsymbol{z}_j)\|\right), \qquad (1)$$

where $\sigma(\cdot)$ is the sigmoid function. The latent positions $z_i$ are involved as Euclidean distances, indicating the relationships between nodes. $\boldsymbol{\gamma}_i$ and $\boldsymbol{\delta}_i \in \mathbb{R}^D$ are the activity and popularity factors, the reverse of which represent the tendencies for a node to send and receive an edge, respectively. The global weights $\beta_{out}$ and $\beta_{in}$ are to measure the importance of activity and popularity, respectively, and $\beta_0$ is a bias. To be mentioned, though Eq. (1) is designed for directed graphs, it can be easily degraded to an undirected variant by setting $\boldsymbol{\gamma}_i = \boldsymbol{\delta}_i$ and $\beta_{out} = \beta_{in}$.

We build an HLSM decoder to generate three types of latent variables, i.e. the latent positions $\boldsymbol{z}_i^{(l)} \in \mathbb{R}^{G_l}$, the community membership $\boldsymbol{s}_i^{(l)} \in \mathbb{R}^{G_l}$ and the node random factors $\boldsymbol{\gamma}_i, \boldsymbol{\delta}_i^{(l)} \in \mathbb{R}^{G_l}$, where $G_l$ is the size of the $l-$th decoder layer. All of these variables are randomly sampled from the variational distributions. At the first layer ($l = 1$), the parameters of variational distributions are defined by priors solely, while for other layers ($l = 2, \ldots, L - 1$), the parameters are obtained by the feedforward representations generated from previous layers as well as priors.

**Latent position**    The latent positions $\boldsymbol{z}_i^{(l)}$ represent the location of each node in a latent space, generated as

$$\boldsymbol{z}_i^{(l)} \sim \mathrm{Normal}\left(\boldsymbol{s}_i^{(l)}\odot f(\mathbf{W}_z^{(l-1)}\boldsymbol{z}_i^{(l-1)}),\mathrm{diag}(\boldsymbol{\sigma}_i^{(l)^2})\right), \qquad (2)$$

where $f(\cdot)$ is a nonlinear activation function (e.g. leaky ReLU), $\odot$ denotes the element-wise multiplication, $\mathrm{diag}(\cdot)$ denotes a diagonal matrix, $\boldsymbol{\sigma}_i^{(l)^2} \in \mathbb{R}^{G_l}$ is the prior variance to be specified, and $\mathbf{W}_z^{(l-1)} \in \mathbb{R}^{G_l \times G_{l-1}}$ is a weight matrix to transform the variable from dimension $G_{l-1}$ to $G_l$.

**Community membership**    The latent positions in Eq. (2) are separated into different communities by the sparse binary variables $\boldsymbol{s}_i^{(l)} = (s_{i1}, \ldots, s_{iG_l})'$, thus an overlapping community structure can

be adapted. Referring to Miller et al. (2009), we employ the Indian buffet process (IBP) prior on $s_i^{(l)}$ to learn the effective number of communities given a sufficiently large truncation parameter $G_l$, i.e.

$$\pi_{ig}^{(l)} = \text{logit}\left(\prod_{h=1}^{g} v_{ih}^{(l)}\right), s_{ig}^{(l)} \sim \text{Bernoulli}\left(\sigma(\pi_{ig}^{(l)})\right). \tag{3}$$

Typically, the log odds $\pi_{ig}^{(l)}$ is generated using a stick-breaking construction, where $v_{ih}^{(l)}$ is drawn from a Beta distribution (Teh et al., 2007). In our model we simplify such hierarchical prior structure by specifying a global $v$ for all nodes. At each layer, the community membership $s_{ig}^{(l)}$ denotes whether node $i$ belongs to community $g$, hence the size of each decoder layer $G_l$ can be explained as the number of communities. Additionally, the proposed HVAE architecture enables our model to detect community structures at multiple levels of granularities. Letting the layer sizes $G_l$ be downward increasing, the top layers indicate the coarse-grained communities and the bottom layers indicate the fine-grained communities.

**Node random factors** To model the prevalent power-law of node degrees, we propose the Dirichlet random factors $\gamma_i^{(l)}, \delta_i^{(l)} \in \mathbb{R}^{G_l}$, i.e.

$$\gamma_i^{(l)} \sim \text{Dirichlet}\left(\xi_i^{(l)} + s_i^{(l)} \odot f(\mathbf{W}_\gamma^{(l-1)}\gamma_i^{(l-1)})\right), \tag{4}$$

$$\delta_i^{(l)} \sim \text{Dirichlet}\left(\psi_i^{(l)} + s_i^{(l)} \odot f(\mathbf{W}_\delta^{(l-1)}\delta_i^{(l-1)})\right), \tag{5}$$

where $\xi_i^{(l)}$ and $\psi_i^{(l)}$ are prior parameters to be specified, and $\mathbf{W}_\gamma^{(l-1)}, \mathbf{W}_\delta^{(l-1)} \in \mathbb{R}^{G_l \times G_{l-1}}$ are weight matrices. Note that the node random factors are dependent on the community membership $s_i^{(l)}$ as well, which can be explained as only the random effects within the community are involved in the representations.

At the output layer ($l = L$), the latent positions $z_i$ and node random factors $\gamma_i$, $\delta_i$ are obtained by linear transformation from dimension $G_{L-1}$ to $D$. Each column of the weight matrices belongs to a $(D-1)$-simplex. Such transformation also changes the interpretation of the layer size, from the number of communities to the dimension of the latent space. Last, the adjacency matrix is reconstructed with the latent variables by Eq. (1).

## 3.2 GCN ENCODER

The proposed DLSM employs a deep encoder as a non-iterative recognition model to infer the parameters of posterior distributions. Assuming the mean-field approximation of the variational distributions, the true joint posterior of the latent variables $p_\theta(\Theta|\mathbf{A}, \mathbf{X})$ can be approximated by a variational posterior $q_\phi(\Theta)$, where $\theta$ and $\phi$ denote the generative (decoder) and inference (encoder) parameters to be trained, respectively. Then, the variational posterior is given as

$$q_\phi(\mathbf{z}, \mathbf{s}, \gamma, \delta) = \prod_{i=1}^{n} \prod_{l=1}^{L} q_\phi(z_i^{(l)}|h_i^{(l)}, z_i^{(l-1)}) q_\phi(s_i|h_i^{(l)}, \pi_i^{(l-1)})$$

$$q_\phi(\gamma_i|h_i^{(l)}, \gamma_i^{(l-1)}) q_\phi(\delta_i|h_i^{(l)}, \delta_i^{(l-1)}), \tag{6}$$

where $h_i^{(l)} \in \mathbb{R}^{K_l}$ is the output of the encoder and $K_l$ denotes the size of the $l$-th layer.

GCN has been proved effective in learning the topology of non-Euclidean data, and thus is an ideal choice for the encoder of our model. Referring to Kipf & Welling (2017), we propose the directed GCN operator as

$$\mathbf{H}^{(l+1)} = f\left(\tilde{\mathbf{D}}_{out}\tilde{\mathbf{A}}\tilde{\mathbf{D}}_{in}\mathbf{H}^{(l)}\mathbf{W}^{(l)}\right). \tag{7}$$

Here $\tilde{\mathbf{A}} = \mathbf{A} + \mathbf{I}_n$ and $\mathbf{I}_n$ is the $n$-dimensional identity matrix. $\tilde{\mathbf{D}}_{out}$ and $\tilde{\mathbf{D}}_{in}$ are diagonal matrices with elements as the out- and in-degrees of $\tilde{\mathbf{A}}$, respectively. $\mathbf{H}^{(0)} = \mathbf{X}$ if $\mathbf{X}$ is available and $\mathbf{H}^{(0)} = \mathbf{I}_n$ if not. During inference, the hidden states are passed to the corresponding layer of the decoder and then combined with the prior information from previous layers to generate the parameters of variational distributions. Note that the vanilla GCN used here can be substituted by any other GNN for directed graphs.

## 4 INFERENCE

We now introduce our fast inference method using the SGVB algorithm (Kingma & Welling, 2014a). Compared with the iterative methods such as MCMC adopted by traditional Bayesian random graph approaches, SGVB is much more efficient and scalable. Such method requires differential Monte Carlo expectations to perform backpropagation, thus the reparameterization trick for each of the latent variables is leveraged.

Denoting $\boldsymbol{z}^* \in \mathbb{R}^{G_l}$ a vector with standard Normal elements, the latent positions are reparametrized as $\boldsymbol{z}_i^{(l)} = \hat{\boldsymbol{\mu}}_i^{(l)}(\boldsymbol{z}_i^{(l-1)}, \boldsymbol{s}_i^{(l-1)}, \boldsymbol{h}_i^{(l)}) + \hat{\boldsymbol{\sigma}}_i^{(l)}(\boldsymbol{z}_i^{(l-1)}, \boldsymbol{s}_i^{(l-1)}, \boldsymbol{h}_i^{(l)}) \odot \boldsymbol{z}^*$, where $\hat{\boldsymbol{\mu}}_i^{(l)}(\boldsymbol{z}_i^{(l-1)}, \boldsymbol{s}_i^{(l-1)}, \boldsymbol{h}_i^{(l)})$, $\hat{\boldsymbol{\sigma}}_i^{(l)}(\boldsymbol{z}_i^{(l-1)}, \boldsymbol{s}_i^{(l-1)}, \boldsymbol{h}_i^{(l)})$ are variational posterior parameters.

Following Mehta et al. (2019), the Bernoulli posterior $p_\theta(s_{ig}^{(l)})$ is approximated by the Binary Concrete distribution (Maddison et al., 2017), i.e.

$$\tilde{s}_{ig}^{(l)} = \frac{1}{\lambda} \left( \hat{\pi}_{ig}^{(l)}(\boldsymbol{s}_i^{(l-1)}, \boldsymbol{h}_i^{(l)}) + \text{logit}(u) \right), \tag{8}$$

where $u \sim U(0,1)$, $\hat{\pi}_{ig}^{(l)}(\boldsymbol{s}_i^{(l-1)}, \boldsymbol{h}_i^{(l)})$ is a variational posterior parameter, $\lambda \in \mathbb{R}^+$ is the temperature to be specified, and then $s_{ig}^{(l)} = \sigma(\tilde{s}_{ig}^{(l)})$.

Referring to the recent literature (Joo et al., 2020), we use multiple normalized Gamma variables with unified rate parameters to compose the Dirichlet distributions $p_\theta(\boldsymbol{\gamma}_i^{(l)})$ and $p_\theta(\boldsymbol{\delta}_i^{(l)})$, i.e.

$$\tilde{\boldsymbol{\gamma}}_i^{(l)} \sim \text{Gamma}\left( \hat{\boldsymbol{\xi}}_i^{(l)}(\boldsymbol{\gamma}_i^{(l-1)}, \boldsymbol{s}_i^{(l-1)}, \boldsymbol{h}_i^{(l)}), \mathbf{1} \right), \tag{9}$$

$$\tilde{\boldsymbol{\delta}}_i^{(l)} \sim \text{Gamma}\left( \hat{\boldsymbol{\psi}}_i^{(l)}(\boldsymbol{\delta}_i^{(l-1)}, \boldsymbol{s}_i^{(l-1)}, \boldsymbol{h}_i^{(l)}), \mathbf{1} \right), \tag{10}$$

where $\hat{\boldsymbol{\xi}}_i^{(l)}(\boldsymbol{\gamma}_i^{(l-1)}, \boldsymbol{s}_i^{(l-1)}, \boldsymbol{h}_i^{(l)})$ and $\hat{\boldsymbol{\psi}}_i^{(l)}(\boldsymbol{\delta}_i^{(l-1)}, \boldsymbol{s}_i^{(l-1)}, \boldsymbol{h}_i^{(l)})$ are variational posterior parameters. For the convenience of notations, here the parameters of Gamma distribution are symbolized as vectors, meaning the element-wise operations. The node random factors are then derived by $\boldsymbol{\gamma}_i^{(l)} = \tilde{\boldsymbol{\gamma}}_i^{(l)} / \sum_j^n \tilde{\boldsymbol{\gamma}}_j^{(l)}$, $\boldsymbol{\delta}_i^{(l)} = \tilde{\boldsymbol{\delta}}_i^{(l)} / \sum_j^n \tilde{\boldsymbol{\delta}}_j^{(l)}$. In practice, the Dirichlet variables are magnified by $n$ times to avoid too small values of $\boldsymbol{\gamma}_i^{(l)}$ and $\boldsymbol{\delta}_i^{(l)}$ when $n$ is large.

In particular, the initial variational posterior parameters ($l = 0$) are set as nonlinear combinations of $\boldsymbol{h}_i^{(1)}$ and $\boldsymbol{h}_i^{(L-1)}$, as illustrated in Fig. 1.

The loss function is defined by minimizing the negative evidence lower bound (ELBO), i.e.

$$\mathcal{L} = \sum_{i=1}^n \sum_{l=1}^L \left( \text{KL}\left[ q_\phi(\boldsymbol{z}_i^{(l)}) \middle\| p_\theta(\boldsymbol{z}_i^{(l)}) \right] + \text{KL}\left[ q_\phi(\boldsymbol{s}_i^{(l)}) \middle\| p_\theta(\boldsymbol{s}_i^{(l)}) \right] + \text{KL}\left[ q_\phi(\boldsymbol{\gamma}_i^{(l)}) \middle\| p_\theta(\boldsymbol{\gamma}_i^{(l)}) \right]$$

$$+ \text{KL}\left[ q_\phi(\boldsymbol{\delta}_i^{(l)}) \middle\| p_\theta(\boldsymbol{\delta}_i^{(l)}) \right] \right) - \sum_{i=1}^n \sum_{j=1}^n \mathbb{E}_q\left[ \log p_\theta(\mathbf{A}_{ij} | \Theta^{(L)}) \right], \tag{11}$$

where $\text{KL}[q(\cdot)||p(\cdot)]$ denotes the Kullback-Leibler (KL) divergence between $q(\cdot)$ and $p(\cdot)$ (of which the full expression is given in Appendix A). The second term is the cross entropy of adjacency matrix reconstruction. Here all of the true posteriors $p_\theta(\Theta^{(l)})$ and variational posteriors $q_\phi(\Theta^{(l)})$, except for the input layer ($l = 1$), are conditioned on $\Theta^{(l-1)}$, which is omitted for simplification.

## 5 MODEL INTERPRETABILITY

In this section, we show that the proposed latent positions and node random factors are interpretable for representing the existence and strength of node influences, by hypothesizing that the true posterior of $\boldsymbol{z}_i$ can be approximated by the variational Normal distribution.

Imagine that there is a new node $i$ with $n_i \geq 2$ neighbors entering a graph and we want to calculate $\boldsymbol{z}_i$ using other existed nodes. It is rational to suppose that the new node should be posited around its

neighbors and thus can be denoted as a residual to the mean of its neighbors, i.e.

$$z_i \sim \text{Normal} \left( \bar{z}_i + \epsilon_i, \text{diag} \left( \sigma_i \right) \right), \tag{12}$$

where $\bar{z}_i = \frac{1}{n_i} \sum_{j \in \mathcal{N}_i} z_j$ is the mean position of node $i$'s neighbors, and $\epsilon_i \in \mathbb{R}^D$ is the residual for $z_i$ to the mean position. Intuitively, the residual should depend on all of the neighbors to various extent, of which the influential ones are likely to contribute more. Formally, we decompose the residual as

$$\epsilon_i = \sum_{j \in \mathcal{N}_i} r_{ij} e_{ij}, \tag{13}$$

where $r_{ij} \geq 0$ is a scalar and $e_{ij} \in \mathbb{R}^D$ is a unit vector at the direction of $z_j - \bar{z}_i$. The non-negative scalar denotes the strength of influences for neighbor $j$ exerting on node $i$ (see Fig. 2 in Appendix for an example in a two-dimensional latent space). An extreme case is $r_{ij} = 0$, which means the latent position of $i$ is not influenced by neighbor $j$ at all.

To further analyze the relation between the latent variables and node influences, we define the prior of $r_{ij}$ as a mixture of a point mass on 0 and an exponential distribution, i.e.

$$f_r(r_{ij}) = \begin{cases} p_0 & \text{if } r_{ij} = 0 \\ (1 - p_0) f_{\text{Exp}_\lambda}(r_{ij}) & \text{if } r_{ij} > 0, \end{cases} \tag{14}$$

where $0 < p_0 < 1$ and $f_{\text{Exp}_\lambda}(\cdot)$ denotes the density function of the exponential distribution with parameter $\lambda$. What is of our interest is the posterior odds ratio (OR) for $r_{ij} > 0$, defined as $\text{OR}(r_{ij} > 0 | \mathbf{A}, \mathbf{X}) = \int_0^\infty f_r(r_{ij} = v | \mathbf{A}, \mathbf{X}) / f_r(r_{ij} = 0 | \mathbf{A}, \mathbf{X}) dv$, which measures the ratio for node influences to exist.

**Theorem 1** *Let $\Theta$ denote a collection of all parameters except $z_i$. The posterior OR for $r_{ij} > 0$ satisfies*

$$OR(r_{ij} > 0 | \mathbf{A}, \mathbf{X}) = \mathbb{E} \left[ g(z_i, \Theta) \Big| r_{ij} = 0, \mathbf{A}, \mathbf{X} \right], \tag{15}$$

The full expression and proof are provided in Appendix B.1. It is interesting that in the proof $(z_i - \bar{z}_i - \sum_{k \in \mathcal{N}_i, k \neq j} r_{ik} e_{ik})' e_{ij}$ is a scalar projection of $z_i$'s residual component onto the unit vector whose direction is determined by $z_j$. In a word, it shows that the posterior latent positions are able to model the existence of node influences via the residual component projections.

Now we consider the interpretation of node random factors, which explains the strengths of node influences. Formally, we have the following conclusion, i.e.

**Theorem 2** *Assume the vectors from $\bar{z}_i$ to all influential neighbors ($r_{ij} > 0$) are orthogonal to each other, and the random factors are the same at all dimensions of the latent space, denoted as $\gamma_i$ and $\delta_i$, respectively. The strength of influences for neighbor $j$ exerting on node $i$ can be denoted as*

$$r_{ij} = h(\gamma_i, \delta_j, \theta, o) \| z_j - z_i \|, \tag{16}$$

*where $\theta$ is the angle between $z_i - \bar{z}_i$ and $z_j - \bar{z}_i$, $o$ is the angle between $\bar{z}_i - z_j$ and $z_i - z_j$.*

Here $h(\gamma_i, \delta_j, \theta, o)$ is a function of the random factors as well as $\theta$ and $o$. The full expression and proof are given in Appendix B.2. Theorem 2 reflects that given the direction of $e_{ij}$ and $z_j - z_i$, the node random factors can be regarded as coefficients for measuring the ratio of $r_{ij}$ to $\| z_j - z_i \|$, which means that the strength of influences for a neighbor delivering to the node can be strengthened or undermined via shrinking or enlarging the distance between them by the random factors.

## 6 EXPERIMENTS

To evaluate the effectiveness and interpretability of the proposed model, we conduct a series of experiments, including link prediction and community detection, on several real-world graphs. Ablation study for the components of our method, including the community membership, node random factors and the HLSM decoder architecture are also conducted.

Table 1: Results (in %) for link prediction on directed graphs. The best results are in **bold** and the second are underlined.

| | Emails | | Political blogs | | Cora | | WikiVote | | Google | |
|---|---|---|---|---|---|---|---|---|---|---|
| | AUC | AP | AUC | AP | AUC | AP | AUC | AP | AUC | AP |
| LSM | $91.3 \pm 0.1$ | $90.6 \pm 0.1$ | $88.7 \pm 0.3$ | $86.5 \pm 0.2$ | $86.2 \pm 0.2$ | $85.7 \pm 0.2$ | N/A | N/A | N/A | N/A |
| VGAE | $86.9 \pm 0.5$ | $85.9 \pm 0.6$ | $84.8 \pm 0.5$ | $83.0 \pm 0.4$ | $72.9 \pm 0.3$ | $77.3 \pm 0.4$ | $78.6 \pm 0.2$ | $77.8 \pm 0.2$ | $78.3 \pm 0.4$ | $80.0 \pm 0.4$ |
| SEAL | $\underline{94.2 \pm 0.3}$ | $\underline{94.0 \pm 0.5}$ | $93.0 \pm 0.3$ | $\underline{93.0 \pm 0.4}$ | $83.2 \pm 0.6$ | $86.8 \pm 0.5$ | $\mathbf{97.1 \pm 0.2}$ | $\underline{96.8 \pm 0.2}$ | $\mathbf{98.8 \pm 0.4}$ | $\underline{98.7 \pm 0.4}$ |
| DGLFRM | $92.4 \pm 0.3$ | $93.1 \pm 0.3$ | $88.6 \pm 0.4$ | $88.4 \pm 0.4$ | $73.7 \pm 0.4$ | $78.3 \pm 0.6$ | $88.4 \pm 0.3$ | $89.8 \pm 0.2$ | $86.0 \pm 0.0$ | $88.2 \pm 0.1$ |
| GGVAE | $91.5 \pm 0.3$ | $90.3 \pm 0.6$ | $\underline{93.6 \pm 0.4}$ | $92.4 \pm 0.5$ | $\mathbf{91.9 \pm 0.8}$ | $\mathbf{92.5 \pm 0.6}$ | $96.2 \pm 0.1$ | $94.7 \pm 0.3$ | $97.8 \pm 0.3$ | $98.2 \pm 0.1$ |
| LGVG | $93.2 \pm 0.2$ | $92.4 \pm 0.3$ | $91.4 \pm 0.4$ | $90.2 \pm 0.3$ | $85.3 \pm 0.2$ | $88.2 \pm 0.4$ | $94.0 \pm 0.0$ | $94.5 \pm 0.0$ | $95.7 \pm 0.0$ | $95.9 \pm 0.0$ |
| DGGAN | $91.4 \pm 0.7$ | $91.0 \pm 0.8$ | $91.2 \pm 0.6$ | $90.8 \pm 0.8$ | $86.1 \pm 0.7$ | $88.6 \pm 0.6$ | $93.2 \pm 0.5$ | $93.4 \pm 0.5$ | $92.7 \pm 0.4$ | $92.8 \pm 0.5$ |
| DLSM | $\mathbf{95.2 \pm 0.2}$ | $\mathbf{94.1 \pm 0.6}$ | $\mathbf{94.8 \pm 0.4}$ | $\mathbf{93.7 \pm 0.8}$ | $\underline{91.6 \pm 0.2}$ | $\mathbf{92.5 \pm 0.2}$ | $\underline{97.0 \pm 0.2}$ | $\mathbf{97.0 \pm 0.2}$ | $\mathbf{98.8 \pm 0.2}$ | $\mathbf{98.8 \pm 0.1}$ |

## 6.1 DATASETS

The experiments of link prediction are conducted on five real-world benchmark datasets, namely Emails (Leskovec et al., 2007), Political blogs (Adamic & Glance, 2005), Cora (Sen et al., 2008), WikiVote (Leskovec et al., 2010) and Google (Palla et al., 2007). All of the graphs have been preprocessed by omitting the isolated nodes and loops. In our experiments, the edges are randomly splitted as 85% for training, 10% for testing, and 5% for validation. More details and descriptive statistics of the datasets are given in Appendix C.

## 6.2 BASELINES

We compare our proposed DLSM[1] with four recent deep generative methods for graph representation learning, i.e. the variational graph auto-encoder (VGAE) (Kipf & Welling, 2016), the deep generative latent feature relational model (DGLFRM) (Mehta et al., 2019), the gravity graph variational auto-encoder (GGVAE) (Salha et al., 2019), the ladder Gamma variational auto-encoder for graphs (LGVG) (Sarkar et al., 2020), and the directed graph generative adversarial network (DGGAN) (Zhu et al., 2021). For link prediction, we also consider a non-generative GNN-based model, i.e. SEAL (Zhang et al., 2021a), which is the state-of-the-art method for this task. The methods that are not particularly designed for directed graphs, including VGAE, DGLFRM and LGVG, are slightly modified by altering the original encoder with our proposed directed GCN given in Eq (7).

## 6.3 EXPERIMENTAL RESULTS

**Link Prediction** For link prediction, we employ the widely used area under the ROC curve (AUC) and average precision (AP) as evaluation metrics. Hyperparameter settings are provided in Appendix D. The experimental results of DLSM and the baselines are presented in Table 1, where the reported results are the means and standard deviations of 10 independent random splits. On all datasets, our model achieves better or comparable performances to the non-generative method SEAL. However, since it requires to materialize a subgraph for each link, SEAL is much more unscalable than our DLSM and takes about 20 multiples longer for running one epoch. The traditional LSM model is unpractical to fit large graphs such as WikiVote and Google because of the MCMC-based inference methods, while the SGVB method adopted by our model is much more computationally efficient. In addition, it seems that GGVAE performs well on approximately unidirectional graphs (i.e. the reciprocal rate is close to 0 as shown in Table 4) such as Cora, whereas on other more generalized directed graphs where more bidirectional edges exist, our method is significantly superior to GGVAE. This is because our proposed DLSM considers the link reciprocity of directed graphs, which is neglected by GGVAE due to its absolutely asymmetric decoding scheme. We also conduct experiments of link prediction on three real-world undirected graphs, given in Appendix E, which show that our proposed model achieves comparable performances with the baselines and verify the capability for modeling undirected graphs as special cases.

**Community Detection** We further conduct community detection on three real-world datasets with ground-truth community labels, i.e. Emails, political blogs and Cora. In particular, for the Emails

---

[1]The source code is available at `https://github.com/upperr/DLSM`.

Table 2: Results (in %) for community detection on directed graphs. The best results are in **bold** and the second are underlined.

| | Emails | | Political blogs | | Cora | |
|---|---|---|---|---|---|---|
| | macro F1 | micro F1 | macro F1 | micro F1 | macro F1 | micro F1 |
| VGAE | $31.5 \pm 1.6$ | $42.5 \pm 1.5$ | $34.9 \pm 1.2$ | $52.3 \pm 1.4$ | $\underline{23.2 \pm 1.0}$ | $\underline{24.8 \pm 0.4}$ |
| DGLFRM | $53.3 \pm 1.3$ | $60.7 \pm 1.0$ | $42.4 \pm 1.1$ | $50.5 \pm 1.3$ | $18.2 \pm 0.5$ | $22.2 \pm 0.5$ |
| GGVAE | $62.8 \pm 1.5$ | $\underline{67.7 \pm 1.4}$ | $\underline{92.0 \pm 1.1}$ | $\underline{92.5 \pm 0.8}$ | $22.7 \pm 0.7$ | $24.0 \pm 0.4$ |
| LGVG | $\underline{67.0 \pm 1.3}$ | $63.8 \pm 1.2$ | $72.8 \pm 1.3$ | $73.9 \pm 1.3$ | $20.0 \pm 0.5$ | $23.9 \pm 0.7$ |
| DGGAN | $62.1 \pm 1.6$ | $66.7 \pm 1.3$ | $88.9 \pm 1.1$ | $89.5 \pm 1.3$ | $18.7 \pm 1.0$ | $21.4 \pm 1.2$ |
| DLSM | $\mathbf{72.0 \pm 1.0}$ | $\mathbf{77.0 \pm 1.2}$ | $\mathbf{92.5 \pm 0.7}$ | $\mathbf{92.6 \pm 0.7}$ | $\mathbf{24.6 \pm 0.8}$ | $\mathbf{25.7 \pm 0.5}$ |

Table 3: Ablation study results (in %) for link prediction.

| | Emails | | Political blogs | | Cora | | WikiVote | | Google | |
|---|---|---|---|---|---|---|---|---|---|---|
| | AUC | AP | AUC | AP | AUC | AP | AUC | AP | AUC | AP |
| -S-$\Delta$-$\Gamma$ | $94.3 \pm 0.4$ | $93.2 \pm 0.4$ | $90.6 \pm 0.3$ | $90.2 \pm 0.4$ | $86.5 \pm 0.5$ | $88.9 \pm 0.6$ | $93.1 \pm 0.4$ | $92.9 \pm 0.3$ | $95.1 \pm 0.3$ | $95.3 \pm 0.4$ |
| -$\Delta$-$\Gamma$ | $93.0 \pm 0.5$ | $91.5 \pm 0.6$ | $90.1 \pm 0.4$ | $87.5 \pm 0.5$ | $82.3 \pm 0.6$ | $86.0 \pm 0.7$ | $87.2 \pm 0.6$ | $85.1 \pm 0.6$ | $94.9 \pm 0.4$ | $95.0 \pm 0.3$ |
| -$\Gamma$ | $94.9 \pm 0.4$ | $93.9 \pm 0.5$ | $91.0 \pm 0.6$ | $88.1 \pm 0.5$ | $85.2 \pm 0.8$ | $87.9 \pm 0.7$ | $93.6 \pm 0.6$ | $93.6 \pm 0.5$ | $98.6 \pm 0.5$ | $98.5 \pm 0.1$ |
| -S | $94.8 \pm 0.5$ | $93.7 \pm 0.6$ | $93.6 \pm 0.5$ | $91.9 \pm 0.6$ | $88.2 \pm 0.6$ | $90.9 \pm 0.8$ | $96.5 \pm 0.3$ | $96.4 \pm 0.3$ | $98.4 \pm 0.2$ | $98.2 \pm 0.1$ |
| -HLSM | $95.2 \pm 0.5$ | $94.0 \pm 0.6$ | $93.9 \pm 0.4$ | $92.5 \pm 0.5$ | $86.8 \pm 0.7$ | $86.9 \pm 0.6$ | $95.6 \pm 0.3$ | $95.8 \pm 0.4$ | $98.0 \pm 0.3$ | $97.3 \pm 0.2$ |
| DLSM | $95.2 \pm 0.2$ | $94.1 \pm 0.6$ | $94.8 \pm 0.4$ | $93.7 \pm 0.8$ | $91.6 \pm 0.2$ | $92.5 \pm 0.2$ | $97.0 \pm 0.2$ | $97.0 \pm 0.2$ | $98.8 \pm 0.2$ | $98.8 \pm 0.1$ |

dataset, communities (departments) with less than 30 nodes are excluded and finally 10 communities are retained. We utilize the node embeddings learned by each model (the latent positions $z_i$ for DLSM) to conduct K-means clustering with ground-truth community numbers. The commonly used macro and micro F1 scores are leveraged as evaluation metrics, which are computed using the most likely mappings between true and predicted clusters. Table 2 shows that our proposed model significantly outperforms all other baselines for graph representation learning. Such results verify the advantages of leveraging the Euclidean distance rather than inner product to compute the divergences between nodes for community detection. (See Appendix G for visualizations of node embeddings using a 2D t-SNE projectionthe distributions of the reverse node random factors, which shows a consistent power-law with node degrees and offers more interpretability.)

**Ablation study**   To justify the effectiveness of each component in our framework, including the community membership $s_i$, node random factors, $\delta_i$, $\gamma_i$, and the HLSM decoder architecture, we compare the proposed DLSM with several variants of our method, which are named by the corresponding ablated components. For example, -$\Delta$-$\Gamma$ indicates the variant removed node random factors (see Appendix F for more details). The link prediction results are presented in Table 3 and the community detection results are in Table 8 of Appendix. Our DLSM performs best on all datasets for both tasks, verifying that all components are useful under the proposed framework. Interestingly, from the first two lines of Table 3, we find that the community membership seems ineffective when the node random factors are not involved. This reflects that the binary $s_i$ mainly benefits our model by shrinking the redundant latent variables to 0 when there are too many parameters to optimize.

## 7 CONCLUSION

We establish a deep generative model to learn multiple highly interpretable node representations for directed graphs. Our proposed DLSM, comprised of a deep GCN encoder and a HLSM decoder, combines the traditional random graphs models with deep learning based methods to take the complementary advantages of interpretability and scalability. Series of experiments have shown that the model is effective for fitting directed graphs and achieves the state-of-the-art performance on link prediction and community detection. In addition, the interpretable node representations learned by the model can naturally represent both the community structure and degree heterogeneity of complex directed graphs. In the future, we shall extend our model for the more complicated scenes such as weighted or dynamic graphs. While the former with multi-valued edges can be simply achieved, the latter demands for a more efficient method to learn the evolutionary topology of graphs.

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

## A  Loss Functions

The KL divergence between Normal distributions Kingma & Welling (2014b) for $\boldsymbol{z}_i^{(l)}$ is

$$\text{KL}\left[q_\phi\left(\boldsymbol{z}_i^{(l)}\right)\middle\|p_\theta\left(\boldsymbol{z}_i^{(l)}\right)\right] = \sum_{g=1}^{G_l} -\frac{1}{2}\left(1 + \log\left(\hat{\sigma}_{ig}^{(l)}\right)^2 - \left(\hat{\mu}_{ig}^{(l)}\right)^2 - \left(\hat{\sigma}_{ig}^{(l)}\right)^2\right). \tag{17}$$

The KL divergence between Binary Concrete distributions Maddison et al. (2017) for $\boldsymbol{s}_i^{(l)}$ is

$$\text{KL}\left[q_\phi\left(\boldsymbol{s}_i^{(l)}\right)\middle\|p_\theta\left(\boldsymbol{s}_i^{(l)}\right)\right] = \sum_{g=1}^{G_l} \log\frac{\hat{\pi}_{ig}^{(l)}\left(1 + \pi_{ig}^{(l)}e^{-\lambda\tilde{s}_{ig}^{(l)}}\right)^2}{\pi_{ig}^{(l)}\left(1 + \hat{\pi}_{ig}^{(l)}e^{-\lambda\tilde{s}_{ig}^{(l)}}\right)^2}, \tag{18}$$

where $\tilde{s}_{ig}^{(l)}$ is the log odds of $s_{ig}^{(l)}$.

The KL divergence between Gamma distributions Joo et al. (2020) for $\boldsymbol{\gamma}_i^{(l)}$ (the same for $\boldsymbol{\delta}_i^{(l)}$) is

$$\text{KL}\left[q_\phi\left(\boldsymbol{\gamma}_i^{(l)}\right)\middle\|p_\theta\left(\boldsymbol{\gamma}_i^{(l)}\right)\right] = \sum_{g=1}^{G_l} \log\frac{\Gamma\left(\xi_{ig}\right)}{\Gamma\left(\hat{\xi}_{ig}\right)} + \left(\hat{\xi}_{ig} - \xi_{ig}\right)\psi\left(\hat{\xi}_{ig}\right), \tag{19}$$

where $\Gamma(\cdot)$ and $\psi(\cdot)$ denote the Gamma and digamma function, respectively.

## B  Details of Theorems

### B.1  Proof of Theorem 1

For any positive value $v > 0$, the posterior probability density of $r_{ij}$ at $v$ is

$$\begin{aligned}
&f\left(r_{ij} = v\middle|\mathbf{A},\mathbf{X}\right) \\
&= \iint f\left(r_{ij} = v, \boldsymbol{z}_i, \Theta\middle|\mathbf{A},\mathbf{X}\right) d\boldsymbol{z}_i d\Theta \\
&= \iint \frac{f\left(\mathbf{A},\mathbf{X}\middle|\boldsymbol{z}_i,\Theta\right)f\left(r_{ij} = v, \boldsymbol{z}_i,\Theta\right)}{f\left(\mathbf{A},\mathbf{X}\right)} d\boldsymbol{z}_i d\Theta \\
&= \iint \frac{f\left(\mathbf{A},\mathbf{X}\middle|\boldsymbol{z}_i,\Theta\right)f\left(r_{ij} = 0, \boldsymbol{z}_i,\Theta\right)}{f\left(\mathbf{A},\mathbf{X}\right)} \cdot \frac{f\left(r_{ij} = v, \boldsymbol{z}_i,\Theta\right)}{f\left(r_{ij} = 0, \boldsymbol{z}_i,\Theta\right)} d\boldsymbol{z}_i d\Theta \\
&= \iint \frac{f\left(r_{ij} = v, \boldsymbol{z}_i,\Theta\right)}{f\left(r_{ij} = 0, \boldsymbol{z}_i,\Theta\right)} \cdot f\left(r_{ij} = 0, \boldsymbol{z}_i,\Theta\middle|\mathbf{A},\mathbf{X}\right) d\boldsymbol{z}_i d\Theta \\
&= \iint \frac{f\left(\boldsymbol{z}_i\middle|r_{ij} = v,\Theta\right)f\left(r_{ij} = v\right)f\left(\Theta\right)}{f\left(\boldsymbol{z}_i\middle|r_{ij} = 0,\Theta\right)f\left(r_{ij} = 0\right)f\left(\Theta\right)} \\
&\quad\cdot f\left(r_{ij} = 0\middle|\mathbf{A},\mathbf{X}\right)f\left(\boldsymbol{z}_i,\Theta\middle|r_{ij} = 0,\mathbf{A},\mathbf{X}\right) d\boldsymbol{z}_i d\Theta. \tag{20}
\end{aligned}$$

Assume that the true posterior of $\boldsymbol{z}_i$ can be approximated by the variational Normal distribution. Then, according to Eq. (12) and (13), we have

$$
\frac{f\left(\boldsymbol{z}_i \middle| r_{ij}=v, \Theta\right) f\left(r_{ij}=v\right)}{f\left(\boldsymbol{z}_i \middle| r_{ij}=0, \Theta\right) f\left(r_{ij}=0\right)}
$$

$$
=\frac{1-p_0}{p_0} \exp \left\{-\frac{1}{2} \prod_d^D \frac{1}{\sigma_{id}^2}\left[\left(\boldsymbol{z}_i-\bar{\boldsymbol{z}}_i-v\boldsymbol{e}_{ij}-\sum_{k \in \mathcal{N}_i, k \neq j} r_{ik}\boldsymbol{e}_{ik}\right)'\left(\boldsymbol{z}_i-\bar{\boldsymbol{z}}_i-v\boldsymbol{e}_{ij}\right.\right.\right.
$$

$$
\left.\left.\left.-\sum_{k \in \mathcal{N}_i, k \neq j} r_{ik}\boldsymbol{e}_{ik}\right)-\left(\boldsymbol{z}_i-\bar{\boldsymbol{z}}_i-\sum_{k \in \mathcal{N}_i, k \neq j} r_{ik}\boldsymbol{e}_{ik}\right)'\left(\boldsymbol{z}_i-\bar{\boldsymbol{z}}_i-\sum_{k \in \mathcal{N}_i, k \neq j} r_{ik}\boldsymbol{e}_{ik}\right)\right]-\lambda v\right\}
$$

$$
=\frac{1-p_0}{p_0} \exp \left\{-\frac{1}{2} \prod_d^D \frac{1}{\sigma_{id}^2}\left[v^2-2v\left(\boldsymbol{z}_i-\bar{\boldsymbol{z}}_i-\sum_{k \in \mathcal{N}_i, k \neq j} r_{ik}\boldsymbol{e}_{ik}\right)'\boldsymbol{e}_{ij}\right]-\lambda v\right\}
$$

$$
=\frac{1-p_0}{p_0} \exp \left\{-\frac{1}{2} \prod_d^D \frac{1}{\sigma_{id}^2}\left[v-\left(\left(\boldsymbol{z}_i-\bar{\boldsymbol{z}}_i-\sum_{k \in \mathcal{N}_i, k \neq j} r_{ik}\boldsymbol{e}_{ik}\right)'\boldsymbol{e}_{ij}-\lambda \prod_d^D \sigma_{id}^2\right)\right]^2\right.
$$

$$
\left.+\frac{1}{2} \prod_d^D \frac{1}{\sigma_{id}^2}\left[\left(\boldsymbol{z}_i-\bar{\boldsymbol{z}}_i-\sum_{k \in \mathcal{N}_i, k \neq j} r_{ik}\boldsymbol{e}_{ik}\right)'\boldsymbol{e}_{ij}-\lambda \prod_d^D \sigma_{id}^2\right]^2\right\}.
$$

By dividing $f(r_{ij}=0|\mathbf{A}, \mathbf{X})$ on both sides of Eq. (20) and integrating over $v$ on $(0, +\infty)$, we can obtain

$$
\mathrm{OR}(r_{ij}>0|\mathbf{A}, \mathbf{X})
$$

$$
=\iint \frac{f\left(\boldsymbol{z}_i \middle| r_{ij}=v, \Theta\right) f\left(r_{ij}=v\right)}{f\left(\boldsymbol{z}_i \middle| r_{ij}=0, \Theta\right) f\left(r_{ij}=0\right)} \cdot f\left(\boldsymbol{z}_i, \Theta \middle| r_{ij}=0, \mathbf{A}, \mathbf{X}\right) d\boldsymbol{z}_i d\Theta
$$

$$
=\iint g\left(\boldsymbol{z}_i, \Theta\right) f\left(\boldsymbol{z}_i, \Theta \middle| r_{ij}=0, \mathbf{A}, \mathbf{X}\right) d\boldsymbol{z}_i d\Theta
$$

$$
=\mathbb{E}\left[g\left(\boldsymbol{z}_i, \Theta\right) \middle| r_{ij}=0, \mathbf{A}, \mathbf{X}\right],
$$

where

$$
g\left(\boldsymbol{z}_i, \Theta\right)=\frac{(1-p_0)\Phi\left(\prod_d^D \sigma_{id}^{-1}\left[\left(\boldsymbol{z}_i-\bar{\boldsymbol{z}}_i-\sum_{k \in \mathcal{N}_i, k \neq j} r_{ik}\boldsymbol{e}_{ik}\right)'\boldsymbol{e}_{ij}-\lambda \prod_d^D \sigma_{id}^2\right]\right)}{p_0\phi\left(\prod_d^D \sigma_{id}^{-1}\left[\left(\boldsymbol{z}_i-\bar{\boldsymbol{z}}_i-\sum_{k \in \mathcal{N}_i, k \neq j} r_{ik}\boldsymbol{e}_{ik}\right)'\boldsymbol{e}_{ij}-\lambda \prod_d^D \sigma_{id}^2\right]\right)},
$$

and $\Phi(\cdot)$ and $\phi(\cdot)$ denote the cumulative distribution function and probability density function of the standard Normal distribution, respectively.

The conclusion is proved.

### B.2 PROOF OF THEOREM 2

We start our proof by giving two definitions, the standard distance (SD) and actual distance (AD) between node $i$ and neighbor $j$. Consider a simple case, where the influence discrepancies of all neighbors are neglected and the latent position $\boldsymbol{z}_i$ is only dependent on the distances between the node and neighbors (see Fig. 2(b) for an illustration). In this case, the residual $\boldsymbol{\epsilon}_i = \mathbf{0}$ and $\boldsymbol{z}_i$ will coincide with $\bar{\boldsymbol{z}}_i$, thus the distance from neighbor $j$ to node $i$ is equal to $\|\boldsymbol{z}_j - \bar{\boldsymbol{z}}_i\|$, which we refer as the standard distance.

Now we reconsider the influence discrepancies of neighbors upon the node. According to Eq. (1), the actual distance between node $i$ and neighbor $j$ when involving node influences is node influences in

our DLSM are involved by rescaling the standard distance. Assume the vectors from the $\bar{z}_i$ to all influential neighbors ($r_{ij} > 0$) are orthogonal to each other, and the random factors are the same at all dimensions of the latent space, denoted as $\gamma_i$ and $\delta_i$, respectively. Then, the actual distance between $z_i$ and $z_j$ is

$$\text{AD}_{ij} = (\beta_{out}\gamma_i + \beta_{in}\delta_j)\|(z_i - z_j)\|. \tag{21}$$

The coefficients $\beta_{out}\gamma_i + \beta_{in}\delta_j$ enlarge or shrink $\text{SD}_{ij} = \|z_j - \bar{z}_i\|$ to $\text{AD}_{ij}$ using the node activity and popularity factors. Let $\theta \in (0, \pi/2)$ be the angle between $z_i - \bar{z}_i$ and $z_j - \bar{z}_i$, i.e.

$$\theta = \arccos\left(\frac{(z_i - \bar{z}_i)'(\bar{z}_j - \bar{z}_i)}{\|z_i - \bar{z}_i\|\|z_j - \bar{z}_i\|}\right). \tag{22}$$

Since $e_{ij}$ is orthogonal to other residual components, the modulus of $\epsilon_i$ can be denoted as $\|\epsilon_i\| = r_{ij}/\cos\theta$. According to the law of cosines, we have

$$\begin{aligned}
\text{AD}_{ij}^2 &= \text{SD}_{ij}^2 + \|\epsilon_i\|^2 - 2\text{SD}\|\epsilon_i\|\cos\theta \\
&= \text{SD}_{ij}^2 + \frac{r_{ij}^2}{\cos^2\theta} - 2\text{SD}_{ij}r_{ij}.
\end{aligned} \tag{23}$$

By combining Eq. (21) and Eq. (23), we obtain

$$\|z_j - \bar{z}_i\|^2 + \frac{r_{ij}^2}{\cos^2\theta} - 2\|z_j - \bar{z}_i\|r_{ij} = (\beta_{out}\gamma_i + \beta_{in}\delta_j)^2\|(z_i - z_j)\|^2$$

$$\left(\frac{r_{ij}}{\cos\theta} - \cos\theta\|z_j - \bar{z}_i\|\right)^2 = \left[(\beta_{out}\gamma_i + \beta_{in}\delta_j)^2 - \sin^2\theta\right]\|(z_i - z_j)\|^2$$

$$\frac{r_{ij}}{\cos\theta} - \cos\theta\|z_j - \bar{z}_i\| = \pm\left[(\beta_{out}\gamma_i + \beta_{in}\delta_j)^2 - \sin^2\theta\right]^{\frac{1}{2}}\|z_j - z_i\|.$$

Given $\theta \in (0, \pi/2)$ (when $\theta \in (\pi/2, \pi)$, $r_{ij} = 0$), the sign for the right side depends on the positive or negative correlation for $\|\epsilon_i\|$ and AD given $\theta$, which is determined by the angle between $\bar{z}_i - z_j$ and $z_i - z_j$, denoted as

$$\begin{aligned}
o &= \arccos\left(\frac{(\beta_{out}\gamma_i + \beta_{in}\delta_j)(z_i - z_j)'(\bar{z}_i - z_j)}{(\beta_{out}\gamma_i + \beta_{in}\delta_j)\|z_i - z_j\|^2}\right) \\
&= \arccos\left(\frac{\bar{z}_i - z_j}{z_i - z_j}\right).
\end{aligned} \tag{24}$$

The sign is positive when $o \in (0, \pi/2 - \theta)$ and negative when $o \in (\pi/2 - \theta, \pi)$.

In summary, we have the following conclusion, i.e.

$$r_{ij} = h(\gamma_i, \delta_j, \theta, o)\|z_j - z_i\|,$$

where

$$h(\gamma_i, \delta_j, \theta, o) = \begin{cases} \cos^2\theta - \cos\theta\left[(\beta_{out}\gamma_i + \beta_{in}\delta_j)^2 - \sin^2\theta\right]^{\frac{1}{2}}, & \text{if } o \in \left(0, \frac{\pi}{2} - \theta\right) \\ \cos^2\theta + \cos\theta\left[(\beta_{out}\gamma_i + \beta_{in}\delta_j)^2 - \sin^2\theta\right]^{\frac{1}{2}}, & \text{if } o \in \left(\frac{\pi}{2} - \theta, \pi\right), \end{cases}$$

and $\theta$, $o$ is defined by Eq. (22) and Eq. (24), respectively.

The conclusion is proved.

## C DIRECTED GRAPH DATASETS

Here we provide more details about the directed graph datasets employed in our experiments. Emails network consists of the members from 42 departments of a European research institution (Leskovec et al., 2007). An edge exists if a person sends at least one email to another. Political blogs is a well-studied social network composed of U.S. political blog nodes (Adamic & Glance, 2005), which are labeled as "liberal" and "conservative" clusters. Two blog pages are connected if one is referenced by another. Cora is a citation network of scientific publications which are classified into seven

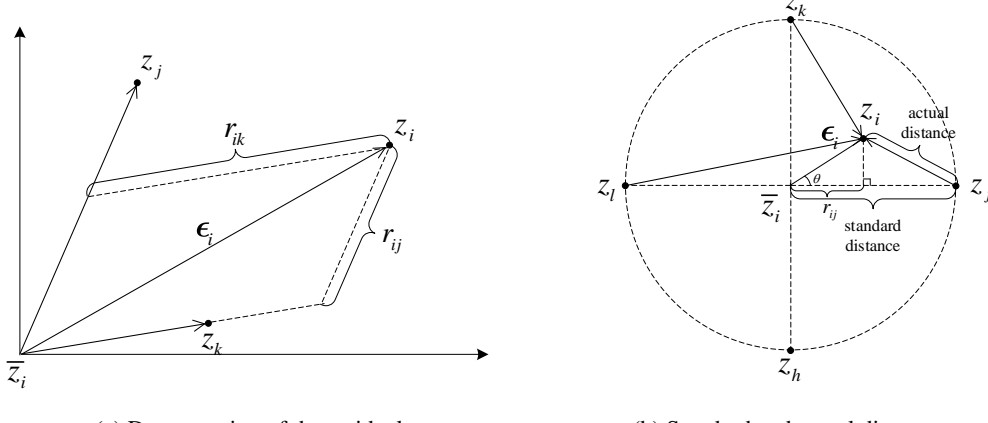

(a) Decomposion of the residual          (b) Standard and actual distances

Figure 2: (a) An illustration for the residual decomposition with two influential neighbors (the influence strengths for other neighbors are assumed to be 0). The residual for $z_i$ can be decomposed as components at the direction of neighbor $z_j$ and $z_k$, and $r_{ij}$, $r_{ik}$ are the corresponding influence strengths. (b) An example for the standard and actual distances in a two-dimensional latent space. The vectors from the mean position $\bar{z}_i$ to influential neighbors, i.e. $z_j$ and $z_k$, are orthogonal and equal length.

Table 4: Descriptive statistics of the real graph datasets. $|\mathcal{V}|$ and $|\mathcal{E}|$ are the numbers of nodes and edges, respectively, CC is the clustering coefficient (Fagiolo, 2007), $d_{max}^{out}$ and $d_{max}^{in}$ are the maximal in-degree and out-degree of all nodes, respectively, $d_{avg}$ is the average degree of all nodes (average in-degree equals to average out-degree), $\text{ED} = \sum_{i=1}^{n} \sum_{j=1}^{n} \mathbf{A}_{ij}/(n(n-1))$ is the edge density, and $\text{RR} = \sum_{i=1}^{n} \sum_{j=1}^{n} \mathbf{A}_{ij} a_{ji} / \sum_{i=1}^{n} \sum_{j=1}^{n} \mathbf{A}_{ij}$ is the reciprocal rate.

| Dataset | $|\mathcal{V}|$ | $|\mathcal{E}|$ | CC | $d_{max}^{out}$ | $d_{max}^{in}$ | $d_{avg}$ | ED | RR |
|---|---|---|---|---|---|---|---|---|
| Emails | 986 | 24,929 | 0.4124 | 333 | 211 | 25.3 | 0.0257 | 0.7112 |
| Political blogs | 1,222 | 19,021 | 0.2459 | 256 | 337 | 15.6 | 0.0127 | 0.2426 |
| Cora | 2,708 | 5,429 | 0.1600 | 5 | 166 | 2.0 | 0.0007 | 0.0556 |
| WikiVote | 7,115 | 103,689 | 0.0896 | 893 | 457 | 14.6 | 0.0020 | 0.0565 |
| Google | 15,763 | 171,206 | 0.4007 | 852 | 11,397 | 10.8 | 0.0007 | 0.2541 |

classes (Sen et al., 2008). Each node and directed edge represent a paper and a citation, respectively. WikiVote is a network of Wikipedia users (Leskovec et al., 2010), where each edge represents a user voting on another to become the administrator. Google is a network of web pages which are connected by hyperlinks (Palla et al., 2007).

The descriptive statistics of the directed graph datasets for link prediction and community detection are summarized in Table 4.

## D    HYPERPARAMETER SETTING AND SENSITIVITY ANALYSIS

The proposed DLSM employs a hierarchical VAE architecture composed of a GCN encoder and HLSM decoder. In practice, both of the two parts have three layers, and the layer sizes are fixed to be 32/64/128 and 50/100/50 for the encoder and decoder, respectively. Note that the last layer of the decoder ($l = L$) is a linear full connection layer which transforms the sparse latent variables to dense for adjacency reconstruction. Fig. 3(a) shows the link prediction results for increasing numbers of layers, and the performance of our method is continuously rising until more than 4 layers.

The HLSM stochastic decoder requires prior distributions to generate latent variables. Specifically, we employ a standard Normal prior for the latent positions $z_i^{(l)}$, i.e. the variance $\sigma_i^{(l)} = 1$. For the community membership $s_i^{(l)}$, we use the IBP prior to infer community numbers given the truncation $G_l$ (i.e. layer sizes of the decoder) and a stick-breaking parameters $v \in (0, 1)$. Typically, a smaller $v$ results in fewer active communities (i.e. non-zero elements of $s_i^{(l)}$). Fig. 3(b) presents the link prediction results of our method with different values of $v$ on the Political blogs dataset, and our method performs best when $v \in (0.9, 1)$. Therefore, we set $v = 0.9$ to ensure the model fully exploring the potential community structure of graphs. Last, we adopt flat priors for the Dirichlet node random factors $\gamma_i$ and $\delta_i$, where all parameters are set as $\xi_{ig}^{(l)} = \psi_{ig}^{(l)} = 1/G_l$.

For inference, the temperature parameter $\lambda$ of the Binary Concrete distribution is fixed to 1. All models are trained by 1,000-2,000 iterations with a learning rate of 0.01 on an RTX 2080 Ti GPU.

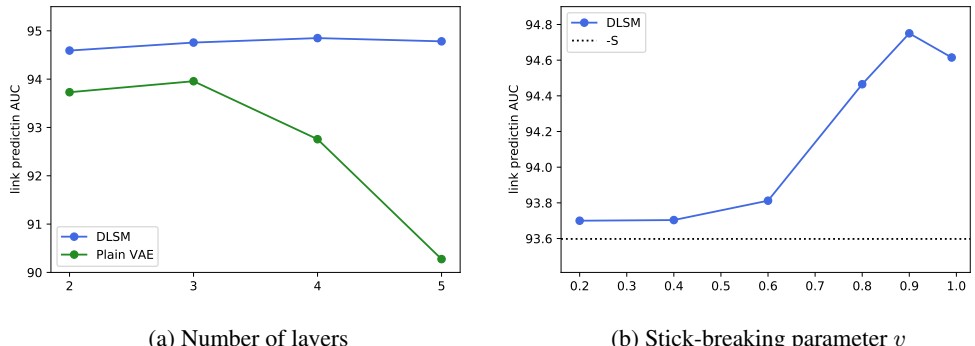

(a) Number of layers          (b) Stick-breaking parameter $v$

Figure 3: Sensitivity analysis for link prediction on the Political blogs dataset. (a) The plain VAE (green) tends to collapse as the number of layers increases, while the performance of DLSM (blue) is much more stable and continuously rises until more than 4 layers. (b) The performance of DLSM (blue solid line) reaches peak when $v$ is about 0.9, and is consistently better than that of the ablated variant -S (black dashed line).

## E    LINK PREDICTION ON UNDIRECTED GRAPHS

Undirected graphs can be regarded as special cases in our framework, as an undirectional edge is treated as a bidirectional one. In this section we provide some experimental results for link prediction on three real-world undirected graph datasets. The baselines are the same as those in the experiments of directed graphs.

### E.1    DATASETS

We consider three real-world undirected graph datasets, namely NIPS12, Cora and Pubmed. NIPS12 is a coauthor network of the authors in NIPS papers from volumes 1-12 Zhou (2015). Cora is the same dataset used in the experiments for directed link prediction, except that the directions of edges are neglected here. Pubmed is a citation network of scientific publications Sen et al. (2008). Both Cora and Pubmed datasets contain sparse bag-of-words feature matrices, which are used as node attributes. The descriptive statistics of the datasets are summarized in Table 5.

### E.2    LINK PREDICTION RESULTS

The results of link prediction on undirected graphs are presented in Table 6. On all datasets, our DLSM achieves the best or comparative performances with the state-of-the-art baselines for undirected graphs.

Table 5: Descriptive statistics of the real-world undirected graph datasets. $|\mathcal{V}|$ and $|\mathcal{E}|$ are the numbers of nodes and edges, respectively, CC is the clustering coefficient, $d_{max}$ is the maximal node degrees, $d_{avg}$ is the average degree of all nodes, ED is the edge density, defined as ED $= 2|\mathcal{E}|/(|\mathcal{V}|(|\mathcal{V}| - 1))$, and FD is the dimension of attribute features (0 denotes no features available).

| Dataset | $|\mathcal{V}|$ | $|\mathcal{E}|$ | CC | $d_{max}$ | $d_{avg}$ | ED | FD |
|---|---|---|---|---|---|---|---|
| NIPS12 | 2,037 | 3,134 | 0.7463 | 45 | 3.1 | 0.0015 | 0 |
| Cora | 2,708 | 5,278 | 0.3893 | 168 | 3.9 | 0.0014 | 1,433 |
| Pubmed | 19,717 | 88,648 | 0.1117 | 171 | 4.5 | 0.0002 | 500 |

Table 6: Results (in %) for link prediction on undirected graphs. The best results are in **bold** and the second are underlined.

| | NIPS12 | | Cora | | Pubmed | |
|---|---|---|---|---|---|---|
| | AUC | AP | AUC | AP | AUC | AP |
| LSM | $85.7 \pm 1.5$ | $86.4 \pm 0.6$ | $86.1 \pm 0.7$ | $87.2 \pm 0.5$ | N/A | N/A |
| VGAE | $87.9 \pm 0.6$ | $91.1 \pm 0.4$ | $92.6 \pm 0.0$ | $93.3 \pm 0.0$ | $94.2 \pm 0.8$ | $93.9 \pm 0.9$ |
| SEAL | $\mathbf{90.7 \pm 0.4}$ | $92.5 \pm 0.5$ | $\mathbf{94.5 \pm 0.6}$ | $\mathbf{95.2 \pm 0.6}$ | $\mathbf{97.0 \pm 0.5}$ | $\underline{96.5 \pm 0.6}$ |
| DGLFRM | $87.3 \pm 0.3$ | $90.1 \pm 0.3$ | $93.4 \pm 0.2$ | $93.8 \pm 0.2$ | $94.0 \pm 0.0$ | $95.0 \pm 0.4$ |
| GGVAE | $89.5 \pm 0.8$ | $90.7 \pm 0.7$ | $89.2 \pm 0.4$ | $90.4 \pm 0.6$ | $93.4 \pm 0.5$ | $94.1 \pm 0.4$ |
| LGVG | $90.1 \pm 0.3$ | $\underline{92.6 \pm 0.7}$ | $\underline{93.6 \pm 0.4}$ | $\underline{95.0 \pm 0.6}$ | $94.6 \pm 0.3$ | $95.6 \pm 0.2$ |
| DGGAN | $89.9 \pm 0.5$ | $91.3 \pm 0.6$ | $93.3 \pm 0.6$ | $94.2 \pm 0.6$ | $94.0 \pm 0.2$ | $94.6 \pm 0.3$ |
| DLSM | $\underline{90.5 \pm 0.1}$ | $\mathbf{92.7 \pm 0.1}$ | $93.1 \pm 0.5$ | $93.5 \pm 0.5$ | $\underline{96.9 \pm 0.2}$ | $\mathbf{96.7 \pm 0.1}$ |

## F   MORE DETAILS FOR ABLATION STUDY

We consider five variants of our proposed model. Here we provide more details about these variants. -HLSM indicates to replace the HLSM decoder of our model using a plain VAE with the same numbers of layers. Others are simplified variants by removing the corresponding components, as presented in Table 7. In particular, $-\Delta$ refers to set $\boldsymbol{\gamma}_i = \boldsymbol{\delta}_i$ (thus the model has degenerated to an undirected form yet still involves degree heterogeneity of graphs), and $-S-\Delta-\Gamma$ can be viewed as a variant of VGAE by changing the plain decoder with our proposed HLSM architecture.

The community detection results of ablation study is given in Table 8. Generally, models including the community membership perform better than those without this component, and the full version DLSM reaches the best performances on all datasets.

## G   VISUALIZATIONS OF NODE REPRESENTATIONS

**Latent position**   We leverage a 2D t-SNE projection (Van der Maaten & Hinton, 2008) to visualize the learned latent positions for the Emails dataset. As comparisons, we also illustrate the node representations learned by DGLFRM and LGVG, both of which also consider the community structure but overlook the degree heterogeneity of graphs. The transformed latent variables learned by these three models are plotted in Fig. 4. It is clearly seen that our model performs best in fitting such directed graphs.

Table 7: Components of ablation variants. A circle denotes the component is involved.

| | HLSM | $z_i$ | $s_i$ | $\delta_i$ | $\gamma_i$ |
|---|---|---|---|---|---|
| $-S-\Delta-\Gamma$ | ○ | ○ | | | |
| $-\Delta-\Gamma$ | ○ | ○ | ○ | | |
| $-\Gamma$ | ○ | ○ | ○ | ○ | |
| $-S$ | ○ | ○ | | ○ | ○ |
| $-HLSM$ | | ○ | ○ | ○ | ○ |
| DLSM | ○ | ○ | ○ | ○ | ○ |

Table 8: Ablation study results (in %) for community detection.

| | Emails | | Political blogs | | Cora | |
|---|---|---|---|---|---|---|
| | macro F1 | micro F1 | macro F1 | micro F1 | macro F1 | micro F1 |
| -S-$\Delta$-$\Gamma$ | $64.8 \pm 1.7$ | $67.8 \pm 1.5$ | $89.6 \pm 0.6$ | $90.7 \pm 0.5$ | $19.0 \pm 0.3$ | $18.9 \pm 0.4$ |
| -$\Delta$-$\Gamma$ | $66.5 \pm 1.2$ | $70.4 \pm 1.2$ | $90.9 \pm 0.6$ | $91.0 \pm 0.6$ | $22.5 \pm 0.5$ | $23.4 \pm 0.4$ |
| -$\Gamma$ | $69.6 \pm 1.3$ | $74.1 \pm 1.2$ | $92.4 \pm 0.4$ | $92.3 \pm 0.5$ | $22.7 \pm 0.6$ | $24.0 \pm 0.5$ |
| -S | $65.3 \pm 1.4$ | $68.5 \pm 1.5$ | $91.1 \pm 0.5$ | $91.3 \pm 0.7$ | $20.9 \pm 0.3$ | $21.6 \pm 0.4$ |
| -HLSM | $70.0 \pm 1.1$ | $72.4 \pm 1.2$ | $90.6 \pm 0.6$ | $90.1 \pm 0.6$ | $23.5 \pm 0.4$ | $24.6 \pm 0.5$ |
| DLSM | $72.0 \pm 1.0$ | $77.0 \pm 1.2$ | $92.5 \pm 0.7$ | $92.6 \pm 0.7$ | $24.6 \pm 0.8$ | $25.7 \pm 0.5$ |

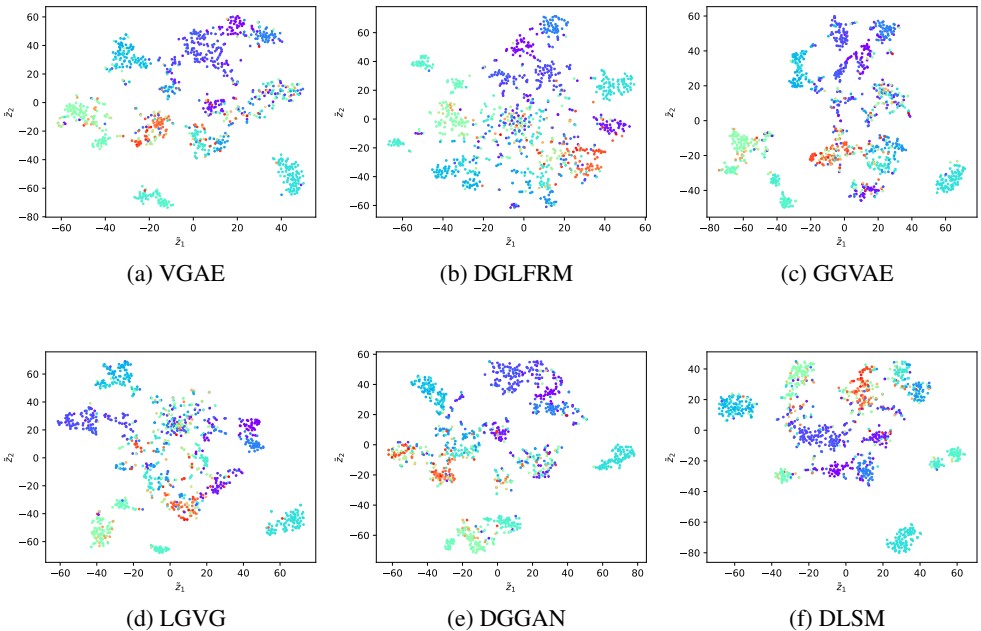

Figure 4: Visualizations of the latent positions learned on the Emails network using a 2D t-SNE projection. Points in different colors denote nodes from the ground-truth communities.

**Community membership** The IBP prior enables the community membership $\boldsymbol{s}_i^{(l)}$ to freely detect the true number of communities. In the experiments we set the truncation $K_l$ to be 50/100, which are far larger than the true numbers of communities. Therefore, the hyperparameter (truncation) actually exert little impact for community detection. In Fig. 5 we illustrate the learned $\boldsymbol{s}_i^{(1)}$ with IBP and independent Bernoulli priors for fitting the Emails network, of which most nodes belong to one of the 10 communities. A lighter color indicate a larger probability for the corresponding dimension (community) of latent variables to be activated ($\neq 0$). It shows that IBP prior condenses the activated communities to be the first 10-15 dimensions, which is relatively approximate to the true number of communities, while the independent Bernoulli variables are much more chaotic and uniformly distributed at all dimensions. Therefore, the proposed IBP prior provide better interpretability for the community membership, in addition to the improvement of model performances.

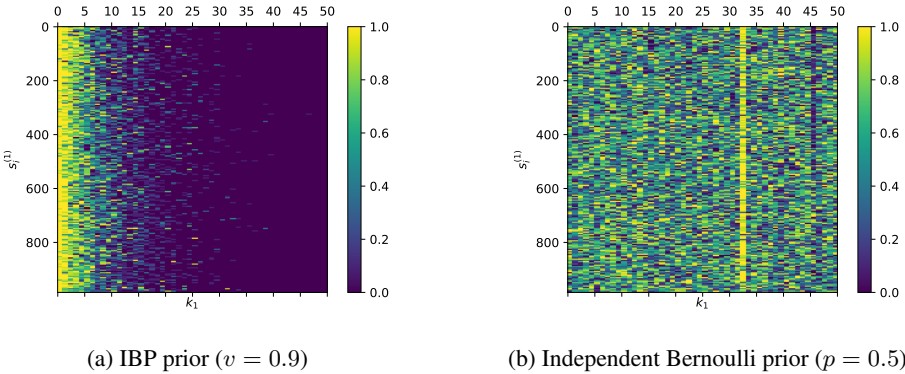

(a) IBP prior ($v = 0.9$)  (b) Independent Bernoulli prior ($p = 0.5$)

Figure 5: Illustration of the community membership with different priors learned on the Emails network.

**Reverse node random factors** The pairwise node random factors $\boldsymbol{\gamma}_i$ and $\boldsymbol{\delta}_i$ are supposed to measure the heterogeneity of out-degrees and in-degrees, respectively, which typically follow the power-law (Barabási & Albert, 1999). Fig. 6(a) and (b) present the probability density distributions (PDD) of the node degrees and reverse random factors learned by DLSM on the political blogs network. It seems that the degree distributions are well fitted by the reverse node random factors, indicating that our DLSM can well represent the degree heterogeneity via these latent variables. Furthermore, Fig. 6(c) illustrates the complementary cumulative distributions (CCD) of the random factors. As can be seen, the logarithm CCD of both $\boldsymbol{\gamma}_i$ and $\boldsymbol{\delta}_i$ are approximately linear, with different slopes though. This shows that the proposed Dirichlet latent variables are flexible enough to accommodate the power-law distribution of degrees.

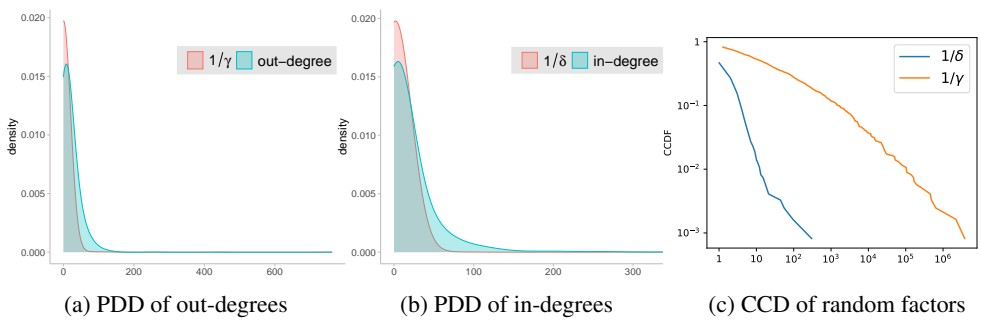

(a) PDD of out-degrees  (b) PDD of in-degrees  (c) CCD of random factors

Figure 6: Probability density distributions of the degrees and node random factors learned by DLSM.

