# OpenReview forum: "A Deep Latent Space Model for Directed Graph Representation Learning"
_ICLR.cc/2022/Conference — ICLR 2022 Submitted_

### Official Review · Reviewer_NCEy · 2021-10-30

**Correctness:** 4
**Technical Novelty And Significance:** 2
**Empirical Novelty And Significance:** 2
**Recommendation:** 6
**Confidence:** 4

**Main Review:**

Pros:
The paper is well-written and easy to read. The details of the model and related derivations are provided.

From a technical viewpoint, the proposed model can be viewed as a new member of the probabilistic directed graph models. Its learning algorithm is based on (amortized) variational inference and the implementation is compatible with neural networks. Overall, the methodology is solid and interpretable.

I like section 5, which provides some useful insights into the model.


Cons:
It seems that the proposed method is not as good as GGVAE on some datasets. It would be nice if the authors can explain this phenomenon to some degree.

In Figure 2, the authors should show the t-SNE figures of GGVAE and VGAE.

Besides node embeddings z’s, the authors introduced another kind of latent code, i.e., s’s to indicate the community structure of a graph. It is worth doing an ablation study to demonstrate the necessity of this latent code.

The influence of the number of layers on the performance of the model should be discussed.


**Summary Of The Paper:**

In this paper, the authors proposed a new directed graph representation model. The proposed model is developed from a Bayesian viewpoint and is implemented in the framework of the variational autoencoder. Additionally, the authors justified the interpretability of the model. They indicate that the proposed model quantitatively captures the influence of each node on its neighbors. Experimental results show that the proposed model performs well in commonly-used datasets.

**Summary Of The Review:**

Overall, the methodology of this work is solid, which is a reasonable combination of several solid probabilistic models. The learning method is easy to implement. Although the novelty of the whole method is not very strong, the experimental results show its feasibility to some degree.

---

> ### Author Response · Authors · 2021-11-21
> **Response to Reviewer NCEy**
>
> Thank you very much for your positive comments!
>
> ### Question 1: experimental results of GGVAE
> * For your first consideration, we have provided some explanations in Section 6.3. Here we will discuss more details. The decoder of GGVAE [1] is given as
>
> $A_{ij}=\sigma(m_j−log\Vert z_i−z_j\Vert^2),$
>
> * where $m_i$ and $z_i$ are learnable variables. As we can see, such decoding scheme is **absolutely asymmetric** ($logit A_{ij}\neq logit A_{ji}$) and thus is specially devised for **unidirectional graphs** where the link reciprocity does not exist. However, our method is much **more generalized** and can model both unidirectional and bidirectional edges of directed graphs. Typically, our method is supposed to well fit any directed graphs, including the approximately unidirectional graphs such as Cora and WikiVote, but it can be more difficult for our method to model these very extreme circumstances. To be mentioned, our method can still significantly outperform GGVAE on some approximately unidirectional graphs like WikiVote, even though it is specially designed for this kind of graphs.
>
> ### Question 2: t-SNE visualization
>
> * Thanks for your advice and we have added the t-SNE visualizations of node embeddings for all baselines in Appendix. The reason why we only consider DGLFRM and LGVG in our original draft is that these two methods have also been verified effective for **community detection** in their own papers [2, 3], and thus are more reasonable and valuable to be compared with our method.
>
> ### Question 3: t-SNE visualization
>
> * Thanks for your advice and we have added the **ablation study** in Section 6 as a justification of the components in our method. It shows that all components, including the latent position, community membership, node random factors and the hierarchical decoder architecture are effective to improve the performances of link prediction and community detection. Particularly, the community membership $s_i$ makes other latent variables to be sparse in our framework. It not only represents the community structure of graphs, but also can **reduce the number of parameters** to be learned and make it easier for our model to be optimized.
>
> ### Question 4: number of layers
>
> * We conduct a **sensitivity analysis** for the number of layers on the political blogs dataset in Appendix D, and the performance of our method continuously increasing until the layers exceeding four.
>
> ### References
>
> * [1] Salha, G., Limnios, S., Hennequin, R., Tran, V. A., and Vazirgiannis, M. (2019). Gravity-inspired graph autoencoders for directed link prediction. In ACM International Conference on Information and Knowledge Management.
> * [2] Mehta, N., Duke, L. C., and Rai, P. (2019). Stochastic blockmodels meet graph neural networks. In International Conference on Machine Learning.
> * [3] Sarkar, A., Mehta, N., and Rai, P. (2020). Graph representation learning via ladder gamma variational autoencoders. In AAAI Conference on Artificial Intelligence.

---

### Official Review · Reviewer_s9T6 · 2021-11-01

**Correctness:** 2
**Technical Novelty And Significance:** 2
**Empirical Novelty And Significance:** 2
**Recommendation:** 3
**Confidence:** 4

**Main Review:**

Strengths:
1. Explicit modeling of community structure, latent positions, and random node factors.
2. The asymmetric modeling of link existence thus suitable for directed graphs.
3. The learned node embeddings have some interpretability.

Weaknesses:
1. Experiments did not compare with state-of-the-art non-VAE-based GNN methods, nor discussed them. For example, the state-of-the-art link prediction method, SEAL [1], should be compared. It can work on directed graphs too. Merely from Table 5 and results reported from [2], SEAL achieves better performance on Cora and Pubmed than the proposed method.
2. Some larger datasets should be used to evaluate the scalability as well as the effectiveness for large networks. Currently, the largest dataset used contains 15,763 nodes and 171,206 edges. However, people are switching gradually to larger datasets with millions of nodes/edges such as Open Graph Benchmark [3].
3. No ablation study on whether the proposed $s,\gamma,\delta$ are useful.

[1] Zhang, Muhan, and Yixin Chen. "Link prediction based on graph neural networks." Advances in Neural Information Processing Systems 31 (2018): 5165-5175.\
[2] Zhu, Zhaocheng, et al. "Neural Bellman-Ford Networks: A General Graph Neural Network Framework for Link Prediction." arXiv preprint arXiv:2106.06935 (2021).\
[3] Hu, Weihua, et al. "Open graph benchmark: Datasets for machine learning on graphs." arXiv preprint arXiv:2005.00687 (2020).

**Summary Of The Paper:**

This paper proposes a VAE-based graph representation learning model for directed graphs. The model is composed of a GCN encoder and a hierarchical latent space model decoder. The decoder explicitly models three kinds of node representations: the node latent position vectors $z$, the community membership vectors $s$ and node random factors $\gamma,\delta$. Specifically, the community membership vectors $s$ are used to gate control which of $z$ and $\gamma,\delta$ pass to the next layer. And finally the adjacency matrix is reconstructed by the distance between $z$'s filtered by node influences $\gamma,\delta$. Experiments show better link prediction and community detection performance than plain VGAE and other VGAE methods.

**Summary Of The Review:**

Overall, although the paper proposes some interesting ideas by explicitly modeling community structure, latent positions, and random node factors, the paper lacks a discussion and comparison with state-of-the-art non-VAE-based baselines and uses relatively small datasets in the experiments, thus is less convincing in performance. Considering the abstract states "The experimental results on real-world graphs demonstrate that our proposed model achieves the state-of-the-art performances on link prediction and community detection tasks", I cannot recommend an accept given that stronger baselines clearly exist.

---

> ### Author Response · Authors · 2021-11-20
> **Response to Reviewer s9T6 (3)**
>
> ### Question 3: ablation study
>
> * We have added the ablation study in Section 6 as a justification of the components in our method. It shows that all components, including the latent position, community membership, node random factors and the hierarchical decoder architecture are effective to improve the performances of link prediction and community detection.
>
> ### References
>
> * [1] Zhang, M., and Chen, Y. (2018). Link prediction based on graph neural networks. Advances in Neural Information Processing Systems, 31, 5165-5175.
> * [2] Hamilton, W. L., Ying, R., and Leskovec, J. (2017). Inductive representation learning on large graphs. In International Conference on Neural Information Processing Systems.
> * [3] Veli?kovi?, P., Cucurull, G., Casanova, A., Romero, A., Lio, P., and Bengio, Y. (2018). Graph attention networks. In International Conference on Learning Representations.
> * [4] Zeng, H., Zhou, H., Srivastava, A., Kannan, R., and Prasanna, V. (2020). Graphsaint: Graph sampling based inductive learning method. In International Conference on Learning Representations.
> * [5] Sarkar, A., Mehta, N., and Rai, P. (2020). Graph representation learning via ladder gamma variational autoencoders. In AAAI Conference on Artificial Intelligence.
> * [6] Funke, T., Guo, T., Lancic, A., and Antulov-Fantulin, N. (2020). Statistical manifold embedding for directed graphs. In International Conference on Learning Representations.
> * [7] Wang, P., Agarwal, K., Ham, C., Choudhury, S., and Reddy, C. K. (2021). Self-supervised learning of contextual embeddings for link prediction in heterogeneous networks. In Web Conference.
> * [8] Zhang, X., He, Y., Brugnone, N., Perlmutter, M., & Hirn, M. (2021). MagNet: A Neural Network for Directed Graphs. arXiv preprint arXiv:2102.11391.

---

> > ### Comment · Reviewer_s9T6 · 2021-11-30
> > **Response to authors**
> >
> > Thanks the authors for the response. I looked into the presented new results. The improvement over SEAL on directed graphs (Table 1) is marginal, and the comparison with SEAL on undirected graphs (Table 6) shows worse performance of the proposed method. Although the authors motivate that the proposed method can learn node representations beneficial for downstream tasks, there is still only a community detection task without the more important node classification task. Further, evaluation on large scale datasets is still missing even after the response. Therefore, I tend to keep my rejection suggestion.

---

> ### Author Response · Authors · 2021-11-20
> **Response to Reviewer s9T6 (1-2)**
>
> Thanks for your comments.
>
> ### Question 1: compare with SEAL
>
> * We consider your suggestion and have added SEAL [1] in the link prediction experiments. The results are presented in Table 1, 2 and 6 of our current draft. It seems that SEAL performs close to our proposed method on some datasets. However, compared to our method, SEAL is more **unscalable** with high time complexity as it requires to materialize a subgraph for each link. (Actually, it takes about 20 multiples longer than our method for running one epoch.)
>
> * More importantly, SEAL and other non-generative models for link prediction cannot generate representations as **node embeddings** for other downstream tasks such as community detection and node classification. There are some gaps between these methods and ours (as well as the baselines selected in our original draft). The **main point** and contribution of our work is to propose a novel graph representation learning method and **generate interpretable latent variables** to better represent the common properties of directed graphs, including link reciprocity, degree heterogeneity and community structure. These learned node representations are readily to be applied for multiple downstream tasks, including but not limited to link prediction and community detection. The experiments in our paper are conducted for the purpose of testing whether these learned representations are able to capture the interpreted graph properties. For example, the link prediction task verifies the ability to represent link reciprocity of directed graphs, and the community detection task verifies the ability for modeling clustering effects.
>
> * We sincerely hope the reviewer to recosider on this point since our work is not acturally aiming for link prediction. Besides, by comparing our method with other graph representation learning methods, we conclude that our method achieves the state-of-the-art performances on downstream tasks.
>
> ### Question 2: evaluation on large-scale graphs
>
> * For the node-based tasks such as node classification, our model is undoubtedly able to handle large-scale graphs since the recognition model leveraged in our model (GCN) can be substituted by any other GNNs for large graphs, such as GraphSAGE [2], GAT [3] and GraphSAINT [4]. However, we have not validated our method for **node classification** in the current draft because it is not the principal downstream task for directed graph representation learning. In the future, we are looking forward to improve our released code and apply our method for node classification on large-scale graphs.
>
> * For link prediction, however, it is still a challenging issue to perform graph representation learning based methods on large-scale graphs due to the difficulty for minibatch sampling. Most previous methods for link prediction on large-scale graphs are heuristic or representative for multiple nodes (e.g. SEAL [1] learns a representation of a subgraph for each link), both of which do not generate embeddings for a **single node** and thus cannot be applied for node-based downstream tasks. Therefore, it is still a developing research field to apply graph representation learning methods for link prediction on large-scale graphs, and most current work just validate their methods on regular graphs with **thousands of nodes** (see [5-8] for several examples). As it is not the main point of this work, we still follow this tradition and evaluate our method on the original datasets for now.

---

### Official Review · Reviewer_bcyE · 2021-11-02

**Correctness:** 3
**Technical Novelty And Significance:** 2
**Empirical Novelty And Significance:** 3
**Recommendation:** 6
**Confidence:** 3

**Main Review:**

Overall this paper is well-written and easy to follow, but the technical contribution is incremental, given that VAEs have been used to represent undirected graph-structured data [1,2,3]. It seems a natural extension of hierarchical VAEs to directed graphs, with some sort of variables for edge direction. The strong points include (1) a VAE with some well-tailed designs to directed graphs, (2) efforts to interpret latent variables, and (3) seemingly promising experimental performance.

The technical weakness mainly comes from the design of the model architecture and experiments. (1) I’m not fully convinced by the model design that variables in the decoder (e.g., z^(i) ) are lay-by-layer influenced by those in the encoder (e.g., H_z^(i)). I’d like to see an intuitive explanation and experiments that show z^(i) conditioned on H_z^(i) performs better than alternative model architectures. (2) As there are so many latent variables to be inferred (n*L*4), seen in Equation 11, I doubt the effectiveness of VI. Is there any model collapse when you optimize Equation 11? You can refer to [4,5,6] for mode collapse in VI. (3) In terms of experiments, I would ask the authors to try some directed graph networks as baselines, not limited to variational or Bayesian. As the authors said, most baselines are “designed for undirected graphs and are not appropriate for learning on asymmetric adjacency matrices”, and the authors modified these baselines for directed graphs. More commonly-used directed graph models as baselines will be convincing, such as [7,8,9]. (4) It’s good to present a model sensitivity analysis of hyper-parameters.

Minor weakness: (1) the \theta in the second last line on page 3 is claimed to denote the collection of model parameters, but it has been used to denote the collection of latent representations in the first paragraph in section 3. Fix this misuse. (2)In the first sentence of the second paragraph on page 4, the authors said four types of latent variables, but in the first sentence of the second paragraph on page 2, they said three types. (3) In the first sentence on page 7, “A extreme” should be “An extreme”. (4) Misplaced subgraph titles in Figure 3 in Appendix.

[1] Thomas N Kipf and Max Welling. Variational graph auto-encoders. In NIPS Workshop on Bayesian Deep Learning, 2016.

[2] Aditya Grover, Aaron Zweig, and Stefano Ermon. Graphite: Iterative generative modeling of graphs. In International Conference on Machine Learning, pp. 2434–2444, 2019.

[3] Guillaume Salha, Stratis Limnios, Romain Hennequin, Viet-Anh Tran, and Michalis Vazirgiannis. Gravity-inspired graph autoencoders for directed link prediction. In ACM International Conference on Information and Knowledge Management, pp. 589–598, 2019.

[4] Lucas, James, et al. "Understanding posterior collapse in generative latent variable models." (2019).

[5] Deasy, Jacob, Nikola Simidjievski, and Pietro Liò. "Constraining variational inference with geometric jensen-shannon divergence." arXiv preprint arXiv:2006.10599 (2020).

[6] He, Junxian, et al. "Lagging Inference Networks and Posterior Collapse in Variational Autoencoders." International Conference on Learning Representations. 2018.

[7] ​​Zhang, Xitong, et al. "MagNet: A Neural Network for Directed Graphs." arXiv preprint arXiv:2102.11391 (2021).

[8] Ou, Mingdong, et al. "Asymmetric transitivity preserving graph embedding." Proceedings of the 22nd ACM SIGKDD international conference on Knowledge discovery and data mining. 2016.

[9] Funke, Thorben, et al. "Low-dimensional statistical manifold embedding of directed graphs." arXiv preprint arXiv:1905.10227 (2019).


**Summary Of The Paper:**

This paper studies a deep latent space model for directed graph representation. The model is basically a hierarchical variational auto-encoder architecture and some latent variables like social activity and popularity factors of node account for the edge directions. Some latent variables are interpreted as positions in the latent space or community membership. The model, therefore, is claimed to be interpretable and superior in experiments.

----Update after rebuttal

The authors basically answered my questions and I'd like to increase the recommendation score. I still think this submission has incremental technical contributions so my new recommendation score is 6 (marginally above the acceptance threshold).

**Summary Of The Review:**

Given the above content, I think this paper is an okay submission, and I will give boardline rejection. I’m willing to increase the score if the authors (1) intuitively and empirically motivate the layer-by-layer variable dependency in the decoder, (2) analyze the effectiveness of VI and how your method prevents mode collapse, (3) conduct comparative experiments with suggested baselines, and (4) analyze how your model is sensitive to the community number.

---

> ### Author Response · Authors · 2021-11-20
> **Response to Reviewer bcyE (3-4)**
>
> ### Question 3: baselines for directed graphs
>
> * Thanks for your advice and we have supplemented a new baseline, i.e. **SEAL** [3] in the link prediction experiments. The results are presented in Table 1, 2 and 6 of our current draft. It seems that SEAL, which is the state-of-the-art method for link prediction, performs close to our proposed method on some datasets. However, there are two significant **drawbacks** of this method. First, it only focuses on link prediction and **cannot learn node embeddings** for other downstream tasks such as community detection and node classification. Moreover, this method is **unscalable** with high time complexity as it requires to materialize a subgraph for each link. (Actually, it takes about 20 multiples longer than our method for running one epoch.)
>
> * We did not consider the **statistical manifold embedding** (SME) [4] because it is an **unsupervised** method and, as the authors claimed, it cannot be applied for supervised tasks such as link prediction. Moreover, the node embeddings learned by SME are one-dimensional scalars, thus it cannot be applied for community detection using clustering methods, either. Here we present the unsupervised learning results for link prediction of SME for your convenience.
>
> AUC:
>
> |        | Emails | Political blogs | Cora | WikiVote | Google|
> | :-----| :----: | :----: | :----: | :----: | :----: |
> | SME [4] | 89.4 | 89.7 | 61.3 | 94.8 | 80.9 |
> | DLSM (ours) | 95.2 | 94.8 | 91.6 | 97.0 | 98.8 |
>
> AP:
>
> |        | Emails | Political blogs | Cora | WikiVote | Google|
> | :-----| :----: | :----: | :----: | :----: | :----: |
> | SME [5] | 88.2 | 87.7 | 64.8 | 93.7 | 82.9 |
> | DLSM (ours) | 94.1 | 93.7 | 92.5 | 97.0 | 98.8 |
>
> * In addition, HOPE [5] is a relatively old method and has been outperformed by one of our baselines, i.e. GGVAE [6]. We therefore did not add this method in the experiments, either. Last, MagNet [7] is a very recent work and we have not compared our method with it yet due to the time limit. We will update our draft if we could finish the experiments before the deadline.
>
> ### Question 4: sensitivity analysis
>
> * The most important hyperparameter of our method is the **stick-breaking parameter $v$** of the IBP prior for community membership, which affects the ability of the model to detect the number of effective communities. We conduct a sensitivity analysis for $v$ on the political blogs dataset and the result is presented in **Fig. 3(b)**. It shows that our method performs best when $v\in(0.9,1)$, as a larger value of $v$ will ensure the model fully exploring the potential community structure of graphs.
>
> ### Minor weakness
>
> * Thank you very much for your kindly reminding and all mistakes have been fixed.
>
> ### References
>
> * [1] Sønderby, C. K., Raiko, T., Maaløe, L., Sønderby, S. K., and Winther, O. (2016). Ladder variational autoencoders. Advances in Neural Information Processing Systems, 29, 3738-3746.
>
> * [2] Bowman, S. R., Vilnis, L., Vinyals, O., Dai, A. M., Jozefowicz, R., and Bengio, S. (2016). Generating sentences from a continuous space. In SIGNLL Conference on Computational Natural Language Learning.
>
> * [3] Zhang, M., and Chen, Y. (2018). Link prediction based on graph neural networks. Advances in Neural Information Processing Systems, 31, 5165-5175.
>
> * [4] Funke, T., Guo, T., Lancic, A., and Antulov-Fantulin, N. (2020). Low-dimensional statistical manifold embedding of directed graphs. In International Conference on Learning Representations.
>
> * [5] Ou, M., Cui, P., Pei, J., Zhang, Z., and Zhu, W. (2016). Asymmetric transitivity preserving graph embedding. In ACM SIGKDD International Conference on Knowledge Discovery and Data Mining.
>
> * [6] Salha, G., Limnios, S., Hennequin, R., Tran, V. A., and Vazirgiannis, M. (2019). Gravity-inspired graph autoencoders for directed link prediction. In ACM International Conference on Information and Knowledge Management.
>
> * [7] Zhang, X., He, Y., Brugnone, N., Perlmutter, M., & Hirn, M. (2021). MagNet: A Neural Network for Directed Graphs. arXiv preprint arXiv:2102.11391.

---

> ### Author Response · Authors · 2021-11-20
> **Response to Reviewer bcyE (1-2)**
>
> Thanks for your positive and valuable comments.
>
> ### Question 1: hierarchical architecture
>
> * At each layer of our hierarchical VAE architecture, the latent variables in the decoder are directly influenced by the hidden states of the encoder through the parameters of their variational posterior distributions. For example, the parameters for the latent position $z_i^{(l)}$ are the mean $\hat{\mu}_i^{(l)}$ and variance $\hat{\sigma}^{(l)}$ vectors, each of which is determined by $h_i^{(l)}$ from the encoder, as well as $z_i^{(l-1)}$ and $s_i^{(l-1)}$ from the last layer of the decoder, i.e.
>
> $\hat{\mu}_i^{(l)}(z_i^{(l-1)},s_i^{(l-1)},h_i^{(l)})=s_i^{(l-1)}\odot f(Wz_i^{(l-1)}) + f(Wh_i^{(l)})$,
>
> $log\hat{\sigma}_i^{(l)}(z_i^{(l-1)},s_i^{(l-1)},h_i^{(l)})=s_i^{(l-1)}\odot f(Wz_i^{(l-1)}+f(Wh_i^{(l)}),$
>
> * where $f(\cdot)$ is an activation function (e.g. ReLU). The first term serves as **prior information** passed from the last layer, and the second term is the **approximate likelihood** skipped from the encoder. We did not give these details in Section 4 due to the page limit.
>
> * The **skipping connections** between the encoder and decoder are very important in our hierarchical VAE architecture because it prevents the latent variables at top layers of the decoder to **collapse** into the prior. As justification, we add the **ablation study** in Section 6 and compare our method to a plain VAE with a 3-layer MLP decoder. See also in **Fig. 3(a)** for an intuitive comparison of the two methods. The experimental results show that the performance of our proposed VAE architecture is significantly better than the plain VAE and can be continuously improved as the number of layers increasing, which indicates that it should be able to mitigate the problem of model collapse.
>
> ### Question 2: model collapse
>
> * It is very thoughtful for you to bring about this issue. The model collapse problem of VAEs refers that both the variational and true posteriors of latent variables collapse to the priors during the training procedure. In our model, as we build a hierarchical decoder architecture, the latent variables generated at top layers have a strong tendency to collapse.
>
> * In our work, we deal with this issue by adopting the following approaches. Firstly, inspired by the ladder VAE [1], we build **skip connections** between the encoder and decoder and add the approximate likelihood of input to the priors at each layer. Such skip connections endow each stochastic latent variable with a deterministic dependency on the input and thus prevent it from collapsing to the priors. Furthermore, we simply follow [2] and leverage the “**KL cost annealing**” method by adding a weight of the KL term in Eq. (11) which increases from a small value to one during training, called the “warm-up” period. Last, we leverage the **IBP prior** to force the community membership (and so the latent position and random factors) to be sparser, which can reduce the number of parameters to be learned and make it easier for our model to be optimized. We find these approaches can empirically mitigate the problem of model collapse from the experimental results presented in Section 6.

---

### Official Review · Reviewer_D8S8 · 2021-11-05

**Correctness:** 3
**Technical Novelty And Significance:** 2
**Empirical Novelty And Significance:** 2
**Recommendation:** 3
**Confidence:** 3

**Main Review:**


The paper proposed an interesting graphical model for modeling the latent factors in directed graph adjacency matrix generation. The proposed solution that leverages VAE for optimizing the ELBO is also valid. Empirical results on the selected benchmarks seem to be significant as well.

However I have several concerns regarding the current draft.

1. Regarding the correctness of the paper, I’m a bit worried about the claims in Sec 5, saying that “we theoretically prove that the latent variables learned by our proposed method, …, are interpretable, …”. Given the fact that the paper uses a variational method to learn a variational posterior, where the posterior takes a mean-field approximation, there’s no guarantee that the model will learn the true posterior. So no matter how good the ideal posterior is, the framework proposed by the paper is not guaranteed to learn that.

2. I’m not sure if the paper is well motivated. The model design seems to be arbitrary, with three components (latent position, community membership, node random factors) combined in an arbitrary way without justification. For example, why one has to adapt the IBP as a prior for Bayesian nonparametric purposes?  Given that you are doing truncation anyway,  I’m wondering whether you would gain anything using the Bayesian nonparametric for community assignment?

3. The benefit of the deep hierarchical graphical model is not properly justified either. From the appendix it seems that the authors are using a 3-layer decoder by default. To justify the benefit of hierarchical graphical models, one should run the experiment with a 1-layer graphical model, parameterized with a 3-layer neural network.

4. The experiments are conducted on small scaled graphs. How practical is it for the proposed method to be applied on realistic social networks? Having results on the latest benchmarks like OGB would be more convincing.


**Summary Of The Paper:**

This paper proposed a directed graphical model called deep latent space model for modeling the graph adjacency matrix generation process, where the latent variables are organized in a hierarchical way with three meaningful components: “latent position”, “community membership” and “node random factors” for each node. The learning leverages the variational framework originated from VAE, with GCN as the core component for posterior parameterization. Experiments on link prediction and community detection show that the proposed method is able to achieve better empirical results.


**Summary Of The Review:**

The paper lacks a motivation and justification for the proposed model design. The theoretical claim might not be correct. The experimental results are not convincing enough and larger scaled experiments on OGB are needed.

---
# After rebuttal:
I would thank the authors' effort in addressing my concerns. However the concerns like the technical contribution and experimental improvements are not fully resolved, but I do appreciate the authors' effort in making the code accessible which improves the reproducibility in the community. I suggest the authors double check the code provided and make sure it is consistent as well.

---

> ### Author Response · Authors · 2021-11-20
> **Response to Reviewer D8S8 (4)**
>
> ### Question 4: application for large-scale graphs
>
> * The main point and contribution of our work is to propose a novel graph representation learning method and generate interpretable latent variables to better represent the common properties of directed graphs. These learned node representations are readily to be applied for multiple downstream tasks, such as link prediction, community detection and node classification. For the node-based tasks such as **node classification**, our model is undoubtedly able to handle large-scale graphs since the recognition model leveraged in our model (GCN) can be substituted by any other GNNs for large graphs, such as GraphSAGE [4], GAT [5] and GraphSAINT [6]. However, we have not validated our method for node classification in the current draft because it is not the principal downstream task for directed graph representation learning. In the future, we are looking forward to improve our released code and apply our method for node classification on large-scale graphs.
>
> * For link prediction, however, it is still a challenging issue to perform graph representation learning based methods on large-scale graphs due to the difficulty for minibatch sampling. Most previous methods for link prediction on large-scale graphs are heuristic or representative for multiple nodes (e.g. SEAL [7] learns a representation of a subgraph for each link), both of which do not generate embeddings for a **single node** and thus cannot be applied for node-based downstream tasks. Therefore, it is still a developing research field to apply **graph representation learning methods** for link prediction on large-scale graphs, and most current work just validate their methods on regular graphs with **thousands of nodes** (see [1, 8-11] for several examples). As it is not the main point of this work, we still follow this tradition and evaluate our method on the original datasets for now.
>
> ### References
>
> * [1] Grover, A., Zweig, A., and Ermon, S. (2019). Graphite: Iterative generative modeling of graphs. In International conference on machine learning.
>
> * [2] Rezaabad, A. L., Kalantari, R., Vishwanath, S., Zhou, M., and Tamir, J. (2021). Hyperbolic graph embedding with enhanced semi-implicit variational inference. In International Conference on Artificial Intelligence and Statistics.
>
> * [3] Sewell, D. K., and Chen, Y. (2015). Latent space models for dynamic networks. Journal of the American Statistical Association, 110(512), 1646-1657.
>
> * [4] Hamilton, W. L., Ying, R., and Leskovec, J. (2017). Inductive representation learning on large graphs. In International Conference on Neural Information Processing Systems.
>
> * [5] Veličković, P., Cucurull, G., Casanova, A., Romero, A., Lio, P., and Bengio, Y. (2018). Graph attention networks. In International Conference on Learning Representations.
>
> * [6] Zeng, H., Zhou, H., Srivastava, A., Kannan, R., and Prasanna, V. (2020). Graphsaint: Graph sampling based inductive learning method. In International Conference on Learning Representations.
>
> * [7] Zhang, M., and Chen, Y. (2018). Link prediction based on graph neural networks. Advances in Neural Information Processing Systems, 31, 5165-5175.
>
> * [8] Sarkar, A., Mehta, N., and Rai, P. (2020). Graph representation learning via ladder gamma variational autoencoders. In AAAI Conference on Artificial Intelligence.
>
> * [9] Funke, T., Guo, T., Lancic, A., and Antulov-Fantulin, N. (2020). Statistical manifold embedding for directed graphs. In International Conference on Learning Representations.
>
> * [10] Wang, P., Agarwal, K., Ham, C., Choudhury, S., and Reddy, C. K. (2021). Self-supervised learning of contextual embeddings for link prediction in heterogeneous networks. In Web Conference.
>
> * [11] Zhang, X., He, Y., Brugnone, N., Perlmutter, M., & Hirn, M. (2021). MagNet: A Neural Network for Directed Graphs. arXiv preprint arXiv:2102.11391.

---

> ### Author Response · Authors · 2021-11-20
> **Response to Reviewer D8S8 (1-3)**
>
> Thanks for your comments.
>
> ### Question 1: theoretical proof
>
> * It is very thoughtful for you to bring about this issue. Actrually, only Theorem 1 about the interpretability of latent positions which represent the existence of node influences is proved based on the hypothesis that the true posteriors can be approximated by the variational Normal distributions. We did not specifically stress this assumption because it is a **basic** and widely adopted premise for plenty of theoretical work based on variational inference methods, e.g. [1], [2]. However, we consider your suggestion and have declared this premise at the **head of Section 5** in our current draft.
>
> ### Question 2: model motivation
>
> * For your concern about the motivation of our proposed model, our method is strongly motivated and each variable of our method is dedicatedly designed regarding the **properties of directed graphs**. As we have explained in Section 1, the latent position, community membership and node random factors are proposed to model the link reciprocity, community structure and degree heterogeneity, respectively.
>
> * The **IBP prior** for community membership is leveraged mainly for two reasons. Firstly, it allows the model to freely **detect the true number of communities** given a sufficiently large truncation. This is very important since the hyperparameter (truncation) can actually exert little impact to the model for community detection. In Appendix D we provide the hyperparameter setting and the truncations are set as 50/100 in the experiments, which are far larger than the true numbers of communities (less than 10 for all datasets). However, our method still performs well for link prediction and community detection by shrinking the variables of redundant communities to 0. Secondly, compared to multiple independent Bernoulli priors, the IBP prior forces the community membership (and so the latent position and random factors) to be sparser, which can dramatically **reduce the number of parameters** to be learned and make it easier for our model to be optimized.
>
> * The **Dirichlet priors** for node random factors, inspired by the classic latent space approaches [3], are also motivated. Specifically, the sparsity of Dirichlet distributions enables our method to model the extreme degree heterogeneity of **scale-free networks**. Meanwhile, the constraint of Dirichlet variables (summing to 1) also addresses the problem of **model recognition** in Eq. (1), namely the likelihood is invariant when the latent position and random factors expand and shrink the same multiple simultaneously.
>
> * We have added the **ablation study** in Section 6 as a justification of the components in our method. It shows that all components, including the latent position, community membership, node random factors and the hierarchical decoder architecture are all effective for the performance improvements of both link prediction and community detection.
>
> * We sincerely hope the reviewer to unbias the judgement on this point, by considering **reviewer bcyE**'s comment that our method is "a VAE with some **well-tailed designs** to directed graphs".
>
> ### Question 3: hierarchical architecture
>
> * We supplement the ablation study in Section 6 and verify the effectiveness of the proposed hierarchical VAE architecture by comparing our model with a **plain VAE** with a 3-layer MLP decoder. The experimental results show that our proposed VAE architecture can significantly improve the model performance and mitigate the problem of **model collapse** in hierarchical VAEs.

---

> > ### Comment · Reviewer_D8S8 · 2021-11-21
> > **RE: Response to Reviewer D8S8**
> >
> > Thanks for your reply and your effort on obtaining more experimental results.
> >
> > I think the additional ablation studies are important and I encourage the authors to include them in the revisions of the paper.
> >
> > I agree that the individual components like the IBP prior or the Dirichlet prior are fundamental and well studied in the literature. However, I'm not convinced that stacking them in such a way is principled. The complication of the model does not necessarily leads to the novelty or technical contribution. There can be exponentially many ways to combine these ingredients, however I'm not convinced that the current combination of them is well motivated for a fundamental problem.
> >
> > I'm not sure what the **graph representation learning methods for link prediction on large-scale graphs** is referring to by the authors. As far as I know, there are many graph representation learning on link prediction works. For example, a recent one like [wang2021relational]
> >
> > > [wang2021relational]  Relational message passing for knowledge graph completion, Wang et.al., KDD 2021

---

> > > ### Author Response · Authors · 2021-11-23
> > > **Further Response to Reviewer D8S8**
> > >
> > > Thank you very much for your new comments.
> > >
> > > **For the motivation of the IBP prior:**
> > >
> > > * The binary community membership is proposed to shrink the redundant variables to 0 and mitigates the “**model collapse**” problem by reducing the parameters to learn. We have conducted experiments by eliminating this variable or change the IBP prior to independent Bernoulli priors. Both of the variants become unstable and tend to collapse when the method runs more than 1000 epochs.
> > > * In addition, it also provides better **interpretability** of our method and make our model to be more robust for hyperparameters. We have added a new part of visualization in Appendix G, which presents the capability for $s$ to detect the true number of communities.
> > >
> > > **For the motivation of the Dirichlet prior:**
> > >
> > > * The node random factors are the critical designation of our framework for directed graphs, since our method will degenerate to an undirected form if they are eliminated. The Dirichlet prior, as we stated in the previous comments, solves the **model recognition** problem, i.e. the likelihood of Eq. (1) is invariant when $\Vert z_i-z_j\Vert$ and $\delta_i$, $\gamma_j$ expand or shrink the same multiple simultaneously. The constraint of Dirichlet variables (summing to 1) prevents $\delta_i$ and $\gamma_j$ increasing to infinity. If we change it to other unrestricted priors such as Normal distributions, the variables will tend to increase to infinity during the training process and the algorithm will break.
> > >
> > > * In a word, both of these two latent variables and their prior distributions are strongly motivated and are indispensable under our framework.
> > >
> > > **For clearer clarification of our method:**
> > >
> > > * What we stated as the "graph representation learning methods" actually means the **deep generative methods** for graph structured data, such as VAE [1], GAN [2] and GraphRNN [3]. These methods focus on generating a full graph (i.e. an adjacency matrix in our method) and usually can be applied for multiple downstream tasks. See [4] for an overview of some recent work. Most current deep generative graph models, however, are unable to be applied for modeling large-scale graphs, especially for the link prediction task, because it requires to generate an entire graph during each run.
> > >
> > > * We have modified the Related Work in our current version and provided a more comprehensive overview of deep generative models for directed and undirected graphs. In addition, we have also modified the contributions of our method for applications on large scale graphs, and the original statement was actually referring to the superiority of our method to the traditional random graph models.
> > >
> > >
> > > * [1] Thomas N Kipf and Max Welling. Variational graph auto-encoders. In NIPS Workshop on Bayesian Deep Learning, 2016.
> > > * [2] Hongwei Wang, Jia Wang, Jialin Wang, Miao Zhao, Weinan Zhang, Fuzheng Zhang, Xing Xie, and Minyi Guo. GraphGAN: Graph representation learning with generative adversarial nets. In AAAI Conference on Artificial Intelligence, volume 32, 2018.
> > > * [3] Jiaxuan You, Rex Ying, Xiang Ren, William Hamilton, and Jure Leskovec. GraphRNN: Generating realistic graphs with deep auto-regressive models. In International Conference on Machine Learning, pp. 5708–5717, 2018.
> > > * [4] Guo, X., & Zhao, L. A systematic survey on deep generative models for graph generation. arXiv preprint arXiv:2007.06686, 2020.

---

> > > > ### Comment · Reviewer_D8S8 · 2021-11-24
> > > > **RE: Further Response**
> > > >
> > > > I would like to thank the authors for the rebuttal and I appreciate your effort.
> > > >
> > > > **Regarding** the modeling, again I'm not asking about the individual component which has already been well justified in the literature. I'm just not in favor of incremental contributions that plug these into the VAE framework for graphs, which by itself didn't address the fundamental issues. An example of fundamental contribution would be the solution for the scalability issue, as the authors pointed out the most existing ones cannot scale up to large graphs. The main benefits of the current paper may come from the two aspects (better accuracy and interpretability), in my understanding. Both of these can be significant if the improvement is well justified, significant, and achieved in an important benchmark. However,
> > > > - regarding the accuracy gain, from the Table 1 it seems the gain is marginal. In many cases it is within the 1-standard deviation of SEAL. With such concerns, I tried the code the author released (thanks for doing that! This indeed improves the transparency during the review process and again I appreciate your effort). It seems the author only release 1 fold of the data, so I can only verify with that portion. What I got is:
> > > > | Dataset | AUC | AP |
> > > > |:----:|:----:|:----:|
> > > > |Emails | 0.936 | 0.925 |
> > > > |Cora | 0.8559 | 0.8925 |
> > > > |Wiki | 0.977 | 0.972 |
> > > >
> > > > It seems the above numbers are much lower than what is reported. Though it is only evaluated on 1 fold, it seems in some cases it is already outside of 1-std of what is reported. The results may vary based on different hyperparameters or splits, but overall it gives the impression that the method is sensitive to parameter tuning and the empirical gain is not significant.
> > > > - regarding the interpretability, the t-SNE plots are good for demonstration, but hard for objective judgements (this is not the authors' fault, but the difficulty brought by the interpretability justification in general).
> > > > - With the newly proposed benchmarks like OGB, it would be more convincing if the proposed approach can improve the link prediction results on it.
> > > >
> > > > **Regarding** the "graph representation learning methods", I'm not convinced that it is only about the deep generative methods. There is a huge community with different principles for approaching the problem. One example is the contrastive learning based method, like **DeepWalk** based ones, or the huge community of knowledge graph embedding like **transE**, **rotatE** that enables the (typed) link prediction, while also offers the embeddings for the nodes that can be visualized using t-SNE. So I'm not fully convinced that the scalability should be an issue. Anyway, my point is to see the improvements on link prediction tasks from challenging benchmarks like OGB.

---

> > > > > ### Author Response · Authors · 2021-12-01
> > > > > **Further Response to Reviewer D8S8**
> > > > >
> > > > > Thanks for your further response and we appreciate your meticulous work for understanding our paper.
> > > > >
> > > > > The main contributions of our work can be summarized as proposing a novel deep generative model for representation learning of directed graphs, which combines the Bayesian random graph model with deep learning-based methods to improve the performances of downstream tasks of graph analysis, such as link prediction and community detection, while generating interpretable node representations.
> > > > >
> > > > > The **fundamental issue** we aim to address in this paper is to learn better node representations for generalized directed graphs. Existing graph representation learning methods mainly focus on undirected graphs or some special cases of directed graphs. For example, [1] and [2] are designed for unidirectional graphs and neglect the link reciprocity property, and [3] proposes statistical manifold embeddings for unsupervised learning and cannot be applied for supervised tasks such as link prediction and node classification.
> > > > >
> > > > > To fill this research gap, we propose a VAE-based deep generative model by combining the traditional latent space model [4] with GCN. Compared to other GNN-based graph representation learning methods [5-7], our generative model concentrates on the learning of full structural information of directed graphs, and thus our method is less application-related to be applied for multiple downstream tasks including link prediction, community detection and node classification.
> > > > >
> > > > > In addition, our proposed hierarchical VAE architecture improves the performances for graph representation learning (justified by link prediction and community detection tasks). Since the design of our model is motivated by the classic latent space random graph model with ideal statistical properties, we believe these improvements are benefited from the nice interpretability of the representations learned by our method, which are respectively interpretated to model the link reciprocity, degree heterogeneity and community structure properties of directed graphs (verified by theoretical proofs and visualizations).
> > > > >
> > > > > Regarding the problem for **experimental result reproduction**, the discrepancies are mainly caused by some hyperparameter settings. We have updated our code [here](https://github.com/upperr/DLSM) . Please download the new version and run
> > > > >
> > > > > > python train.py --dataset political --use_kl_warmup 1 --epochs 2000 --encoder 32_64_128 --decoder 50_100 --directed 1 --learning_rate 0.01 --split_idx 0 --features 0
> > > > >
> > > > > One of the main differences between the two versions is that we add a “warm up” weight to the KL term in Eq. (11), which increases from a small value to one during training, to mitigate the “model collapse” problem.
> > > > >
> > > > > For justifications of **model interpretability**, as the reviewer said, it is uneasy to provide objective judgements. In our paper, we attempt to verify the interpretability of the proposed representations from both the theoretical and empirical aspects. Firstly, in Section 5, we show that the latent position and node random factors are interpretable for representing the existence and strength of node influences. Secondly, in Appendix G, we visualize the node representations to verify that the latent position, community membership and node random factors can empirically model the corresponding interpreted graph characteristics, namely the position, community and degree of nodes.
> > > > >
> > > > > We thank the reviewers’ suggestions of evaluating on OGB and we consider updating the experimental results in the future. As the reviewer have mentioned, scalability is indeed another challenging issue and remains an open research question of generative methods of graph representation learning.

---

> > > > > ### Author Response · Authors · 2021-12-01
> > > > > **Further Response to Reviewer D8S8**
> > > > >
> > > > > We thank the reviewers’ suggestions of evaluating on OGB and we consider updating the experimental results in the future. As the reviewer have said, scalability is indeed another challenging issue and remains an open research question of generative methods of graph representation learning.
> > > > >
> > > > > We agree that there are plenty of graph representation learning methods other than generative models, such as DeepWalk and GNN-based methods. What we were trying to clarify is that: our proposed method is a deep generative model based on VAE. Please refer to the Related Work in Section 2 for more details about deep generative graph models.
> > > > >
> > > > > [1] Guillaume Salha, Stratis Limnios, Romain Hennequin, Viet-Anh Tran, and Michalis Vazirgiannis. Gravity-inspired graph autoencoders for directed link prediction. In ACM International Conference on Information and Knowledge Management, pp. 589–598, 2019.
> > > > >
> > > > > [2] Xitong Zhang, Yixuan He, Nathan Brugnone, Michael Perlmutter, and Matthew Hirn. Magnet: A neural network for directed graphs. arXiv preprint arXiv:2102.11391, 2021b.
> > > > >
> > > > > [3] Thorben Funke, Tian Guo, Alen Lancic, and Nino Antulov-Fantulin. Statistical manifold embedding for directed graphs. In International Conference on Learning Representations, 2020.
> > > > >
> > > > > [4] Peter D Hoff, Adrian E Raftery, and Mark S Handcock. Latent space approaches to social network analysis. Journal of the American Statistical Association, 97(460):1090–1098, 2002.
> > > > >
> > > > > [5] Thomas N Kipf and Max Welling. Semi-supervised classification with graph convolutional networks. In International Conference on Learning Representations, 2017.
> > > > >
> > > > > [6] Petar Veliˇckovi´c, Guillem Cucurull, Arantxa Casanova, Adriana Romero, Pietro Lio, and Yoshua Bengio. Graph attention networks. In International Conference on Learning Representations, 2018.
> > > > >
> > > > > [7] Muhan Zhang, Pan Li, Yinglong Xia, Kai Wang, and Long Jin. Labeling trick: A theory of using graph neural networks for multi-node representation learning. In International Conference on Neural Information Processing Systems, 2021a.

---

### Decision · Program_Chairs · 2022-01-20

**Decision:**

Reject

**Comment:**

This to me looks like quality work not yet adequately developed, and thus is borderline work.  The authors seem to have achieved a good result:  equalling SotA SEAL (although, one reviewer did preliminary experiments and could not match this) with a sophisticated algorithm using a variety of Bayesian tricks, a more scalable algorithm, and one potentially adapted to further tasks.  However, not all of these impressive feats are adequately demonstrated in this paper, though many had parts included in the rewrite.  So I'd say the paper needs a rewrite and more focussed experimental work to broaden the presentation of empirial performance, for instance to node classification.
I certaintly appreciated the use of IBP and Dirichlet models within the system, so would love to see the work further developed.
The reviewers agreed in several aspects:  (1) more experimental work, for instance on better and larger benchmark data, (2) better presentation and discussion of the theory, (3) better discussion of the motivation for the model (as per reviewer D8S8), and oftentimes linked to the ablation study to support this, which you have done some of (4) additional connections to recent related work in graph representation learning on link prediction works
The authors have done a good job or addressing many of the reviewers concerns, ultimately lifting the paper from Reject to Borderline Negative, but I think more work is needed.